

# EARice10: A 10 m Resolution Annual Rice Distribution Map of East Asia for 2023

Mingyang Song[1,2,3], Lu Xu[1,2], Ji Ge[1,2,3], Hong Zhang[2,1,3*], Lijun Zuo[1,2], Jingling Jiang[1,2,3], Yinhaibin Ding[1,2,3], Yazhe Xie[1,2,3], Fan Wu[1,2]

[1]Key Laboratory of Digital Earth Science, Aerospace Information Research Institute, Chinese Academy of Sciences, Beijing 100094, China
[2]International Research Center of Big Data for Sustainable Development Goals, Beijing 100094, China
[3]College of Resources and Environment, University of Chinese Academy of Sciences, Beijing 100049, China

*Correspondence to*: Hong Zhang (zhanghong@radi.ac.cn)

**Abstract.** Timely and accurate high-resolution annual mapping of rice distribution is essential for food security, greenhouse gas emissions assessment and supporting for sustainable development goals. East Asia (EA), a major global rice-producing region, accounts for approximately 29.3% of the world's rice production. Therefore, to acquire the latest rice distribution of the EA, this study proposed a novel rice distribution mapping method based on the Google Earth Engine (GEE) platform, producing a 10-meter-resolution annual rice distribution map (EARice10) of EA for 2023. A new Synthetic Aperture Radar (SAR)-based Rice distribution Mapping Index (SRMI) was firstly proposed and combined with optical indices to generate representative rice samples. In addition, a stacking-based optical-SAR adaptive fusion model was designed to fully integrate the features of Sentinel-1 and Sentinel-2 data for high-precision rice mapping in EA. The accuracy of EARice10 was evaluated using more than 90,000 validation samples and achieved an overall accuracy of 90.48%, with both user's and producer's accuracies exceeding 90%. The reliability of the product was verified by an $R^2$ values ranging between 0.94 and 0.98 with respect to official statistics, and between 0.79 and 0.98 with respect to previous rice mapping products. EARice10 is accessible at https://doi.org/10.5281/zenodo.13118409 (Song et al., 2024).

## 1 Introduction

Rice is a primary global food source, occupying approximately 11.21% of the world's agricultural land and feeding over half of the global population, according to recent Food and Agriculture Organization (FAO) data (Zhang et al., 2018; Xu et al., 2023; Fao, 2024). As the global population continues to increase, so does the demand for rice in human societies. In 2022, the total rice production in EA reached 227,494,000 tons, which accounted for about 29.3% of the total global rice production (Chen and Zhao, 2023; Fao, 2024). Therefore, timely and accurate mapping of rice distribution in EA is critical for realizing the United Nations Sustainable Development Goal 2 (SDG 2) (Zhang et al., 2018).

Traditional methods of rice area mapping rely heavily on manual surveys, which are often labor-intensive and time-consuming (Pan et al., 2021; Abdali et al., 2023). With the open access to data from remote sensing satellites such as





MODIS, Landsat series, and Sentinel series, remote sensing has become an effective tool for mapping the spatial distribution of rice on a large scale (Dong and Xiao, 2016; Abdali et al., 2023; Gao et al., 2023; Zhang et al., 2023a). Currently, studies have been conducted to produce rice distribution maps in EA using various remote sensing data. For example, Xiao et al. and Han et al. used MODIS data to create 500 m resolution rice area maps covering South China and the Asian monsoon region,

respectively (Xiao et al., 2005a; Han et al., 2022); Carrasco et al. created 30 m resolution rice area maps for Japan from 1985 to 2019 using phenological algorithms and Landsat data (Carrasco et al., 2022); Jo et al. used Recurrent U-Net and Sentinel-1 data to map rice distribution in the Republic of Korea at 10 m resolution from 2017 to 2021 (Jo et al., 2023); Han et al. combined MODIS and Sentinel-1 data to release a 10 m resolution rice distribution dataset, NESEA-Rice10, covering Northeast and Southeast Asia (Han et al., 2021); Pan et al. used Sentinel-1 data and the TWDTW algorithm to map the

distribution of double-cropping rice in nine southern provinces of China (Pan et al., 2021); Shen et al. used the TWDTW algorithm and Sentinel-1/2 data to map the distribution of single-cropping rice across 21 provinces in China (Shen et al., 2023). Existing rice distribution datasets with complete coverage of EA are mainly produced using 500 m resolution MODIS data, while 10 m resolution rice distribution datasets mapped using Sentinel-1/2 data cover only part of EA and still does not fully cover the whole region.

The Google Earth Engine (GEE) cloud platform integrates remote sensing images with varying temporal and spatial resolutions and provides professional image processing and classification algorithms, which has become the main platform for national and subcontinental-scale rice area mapping (Gorelick et al., 2017; Yu et al., 2023). Approaches to map rice area on the GEE platform can be summarized as phenology-based and machine learning-based approaches (Dong et al., 2016; Ni et al., 2021). Phenology-based methods are relatively easy to understand and to practice, which usually determine rice

planting areas with specific spectral bands or vegetation indices that are sensitive to the "water-soil-vegetation" characteristics of rice paddies during specific phenological stages (Xiao et al., 2005b; Xiao et al., 2006; Han et al., 2021; Zhan et al., 2021; Carrasco et al., 2022; Gao et al., 2023; Xu et al., 2023). Machine learning-based approaches depend less on the rice phenological stages by digging the relationship between specific bands or indices and the label information, and to achieve high rice recognition accuracies, a large amount of training data are demanded to train the supervised model, such

as Random Forest (RF) (He et al., 2021; You et al., 2021), and Support Vector Machine (SVM) (Ni et al., 2021; Huang et al., 2023).

The robustness of machine learning models relies on extensive high-quality training samples. Obtaining samples through manual labeling is often costly and time-consuming, especially in large-scale land cover mapping tasks. Some studies proposed to generate sample sets with existing public datasets to support the fine classification of crops (Song et al., 2017;

Hao et al., 2020; Xu et al., 2020; Johnson and Mueller, 2021; Wen et al., 2022; Yang et al., 2023; Pandžić et al., 2024), such as the United States Department of Agriculture's (USDA) Cropland Data Layer (CDL) (Johnson and Mueller, 2010) and the Agriculture and Agri-Food Canada's (AAFC) Crop Inventory (CI) dataset (Fisette et al., 2013). However, the limited availability of moderate to high resolution (10－250 m) rice distribution maps in EA does not allow the use of historical data to generate reliable training samples for the entire region (Lin et al., 2022; Zhi et al., 2022; Sun et al., 2024). Some efforts



have been made to generate training samples automatically using current-season remote sensing data. Zhang et al. generated training sample sets using optical remote sensing indices during multiple specific phenological stages of rice (Zhang et al., 2023b), while Gao et al. combined SAR data and object-oriented methods to generate training samples (Gao et al., 2023). However, when mapping rice distribution in EA, the vast differences in climate conditions and landscape complexity across different areas make it insufficient to rely on a single data source to obtain high-quality training datasets for various regions.

Therefore, a new sample generation strategy is urgently needed to fully utilize both optical and SAR data.

Currently, studies have been conducted to combine optical and SAR data for rice distribution mapping and demonstrated that the classification accuracy of the combined data sources is superior to that of a single data source (Chen et al., 2020; Xiao et al., 2021; Gao et al., 2023; Wang et al., 2024). However, existing studies have mainly used SAR features as complementary features to optical data, and optical and SAR are fed into the same model (Chen et al., 2020; He et al., 2021), which does not

take into account the respective characteristics of SAR and optical data. Therefore, it is necessary to design a new rice classification strategy that can dig the intrinsic features of optical and SAR data independently and effectively integrate their advantages to obtain better classification results.

To address these challenges, this study designs a high-precision rice distribution mapping framework applicable to different regions in EA. The framework takes the advantages of both the phenology-based and machine learning-based approaches: (1)

a novel SAR-based rice mapping index (SRMI) is proposed and combined with optical indices to generate representative training samples; (2) an optical-SAR adaptive fusion model that considers the impacts of cloud cover on the rice recognition is designed, which makes full use of the features of Sentinel-1 and Sentinel-2 data with a stacked model and can achieve accurate rice mapping results. Based on the proposed method, a 10 m resolution rice area map for EA in 2023 (EARice10) is produced, providing essential data for monitoring the growth and estimating the yield of rice in EA.

**2 Materials**

**2.1 Study area**

The EA region includes five countries: China, Japan, the Democratic People's Republic of Korea, the Republic of Korea, and Mongolia. Rice is cultivated in all these countries except Mongolia, as shown in Figure 1-a.

Japan, the Democratic People's Republic of Korea, the Republic of Korea, and China encompass diverse climatic conditions,

landscape features, and agricultural practices, leading to significant regional variations in rice planting areas and cropping systems (Luo et al., 2020; Hu et al., 2023). Based on cropping systems, the entire EA can be categorized into single-season, double-season, and mixed-season areas, where the mixed-season area includes a mixture of single-season and double-season rice (see Figure 1-b).

Single-season rice is the dominant system in the Democratic People's Republic of Korea, the Republic of Korea, and Japan

due to thermal limitations, with transplanting typically occurring from May to June. As the world's largest rice producer, the situation of rice cultivation in China is more complex. There are 31 provincial-level administrative regions in China where

rice can be grown (except for Qinghai Province, Hong Kong, and Macao). Eight of these provinces, Beijing, Tianjin, Hebei, Shanxi, Tibet, Gansu, Ningxia, and Xinjiang, have less than 100,000 hectares (ha) under rice cultivation. Single-season rice is grown in northern China, while double-season rice is prevalent in southern China, with early rice transplanting from
March to April and late rice transplanting from July to August (Pan et al., 2021; You et al., 2021; Shen et al., 2023).

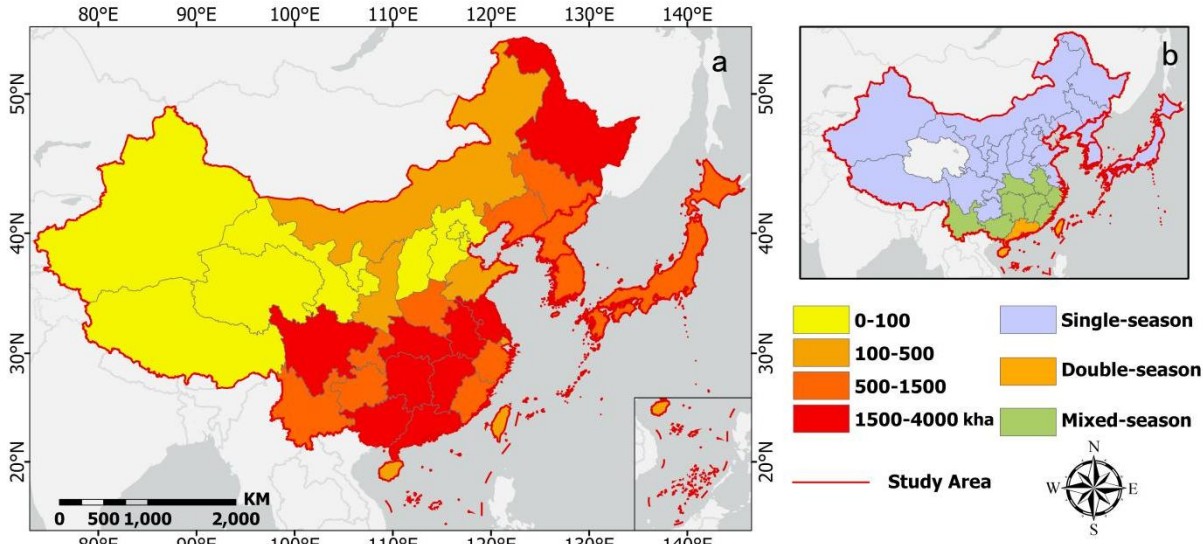

**Figure 1. Overview of the study area: (a) rice planting areas; (b) rice cropping systems.**

## 2.2 Satellite imagery

In this study, Sentinel-1 and Sentinel-2 data were used for rice mapping at 10 m resolution. In 2023, a total of more than
202,000 Sentinel-2 images and over 13,000 Sentinel-1 images covered the study area.

### 2.2.1 Sentinel-2

The Sentinel-2 (S2) mission comprises two twin satellites, S2A and S2B, providing wide-swath, high-resolution, multi-spectral imagery with a global revisit time of 5 days (Zhao et al., 2021), with the Multi-Spectral Instrument (MSI) acquires data in 13 spectral bands, including visible and near-infrared (NIR) bands at 10 m resolution, red-edge and shortwave
infrared (SWIR) bands at 20 m resolution, and atmospheric bands at 60 m resolution. The red-edge bands of S2 have proven particularly valuable for vegetation and agricultural monitoring applications (Griffiths et al., 2019; You et al., 2021; Zhang et al., 2022b). This study used S2 surface reflectance (SR) data (Level-2A), which more accurately reflects ground object information compared to S2 top-of-atmosphere (TOA) data (Level-1C), making it more suitable for rice extraction (Shelestov et al., 2017; Ni et al., 2021).

To leverage the rich spectral information of S2 data, we utilized the 10 original bands and several commonly used spectral indices as input features for the classification model. The original bands include blue, green, red, near-infrared (NIR), red-edge (RE) 1-4, and SWIR 1-2. All bands were resampled to a 10 m spatial resolution. The selected spectral indices,





commonly employed for rice mapping, include the Normalized Difference Vegetation Index (NDVI), Land Surface Water
Index (LSWI), Enhanced Vegetation Index (EVI), Bare Soil Index (BSI), Plant Senescence Reflectance Index (PSRI), and
Green Chlorophyll Vegetation Index (GCVI) (Table 1) (Ni et al., 2021; Gao et al., 2024; Zhu et al., 2024). To maximize data
availability while minimizing cloud contamination, we applied the Cloud Score+ algorithm to all S2 SR images acquired
during the rice growing season (Pasquarella et al., 2023). Subsequently, we generated semi-monthly cloud-free composites
by averaging the cloud-masked pixel values within each composite period. All the pre-processing and feature extraction
were accomplished on the GEE platform.

**Table 1. The spectral indices used in this study.**

| Spectral Index | Formula | Reference |
|:---:|:---:|:---:|
| NDVI | $NDVI = \dfrac{\rho_{NIR} - \rho_{Red}}{\rho_{NIR} + \rho_{Red}}$ | (Pasquarella et al., 2023) |
| LSWI | $LSWI = \dfrac{\rho_{NIR} - \rho_{SWIR1}}{\rho_{NIR} + \rho_{SWIR1}}$ | (Xiao et al., 2004) |
| EVI | $EVI = 2.5 \times \dfrac{\rho_{NIR} - \rho_{Red}}{\rho_{NIR} + 6 \times \rho_{Red} - 7.5 \times \rho_{Blue} + 1}$ | (Huete et al., 1997) |
| BSI | $BSI = \dfrac{(\rho_{SWIR1} + \rho_{Red}) - (\rho_{NIR} + \rho_{Blue})}{(\rho_{SWIR1} + \rho_{Red}) + (\rho_{NIR} + \rho_{Blue})}$ | (Bera et al., 2020) |
| PSRI | $PSRI = \dfrac{\rho_{Red} - \rho_{Blue}}{\rho_{RE2}}$ | (Merzlyak et al., 1999) |
| GCVI | $GCVI = \dfrac{\rho_{NIR}}{\rho_{Green}} - 1$ | (Gitelson et al., 2003) |

**2.2.2 Sentinel-1**

Sentinel-1 (S1) provides C-band (5.405 GHz) SAR data at 10 m spatial resolution with a 12-day revisit time, making it a
valuable data source for various applications, including agricultural monitoring (Wei et al., 2019; Xu et al., 2021; Tian et al.,
2023; Xu et al., 2023).  This study utilized S1 Interferometric Wide Swath (IW) mode Ground Range Detected (GRD) data,
comprising VH and VV polarization channels.

The S1 data on the GEE platform undergoes basic pre-processing (e.g., thermal noise removal, radiometric calibration,
terrain correction). To obtain higher quality SAR features, we further processed the S1 data on GEE using the S1 Analysis
Ready Data (ARD) framework described in (Mullissa et al., 2021). This additional processing involved border noise removal,
speckle filtering using the Refined Lee filter, and radiometric terrain normalization. To generate spatially and temporally
consistent S1 composites, median compositing was performed on all available data within the rice growing season at a 12-
day interval. In addition to the original VH and VV backscattering coefficients, the cross-polarization ratio (CR) defined as
VH/VV was calculated, which has proven valuable for crop classification (Veloso et al., 2017; D'andrimont et al., 2021).



### 2.3 Auxiliary data

Several auxiliary datasets were incorporated, including land cover products, a digital elevation model (DEM), rice phenology
data, existing rice distribution maps, statistical yearbook data, and validation samples (Table 2).

To minimize classification errors associated with non-cropland areas and account for potential inter-annual cropland changes, we applied a cropland mask generated by merging the cropland classes from the ESA WorldCover 2020 and 2021 products (Gao et al., 2023; Wang et al., 2024).

The Shuttle Radar Topography Mission (SRTM) Version 3 digital elevation model (DEM) at 1 arc-second resolution
(approximately 30 m)  was used for S1 data pre-processing and analysis of rice spatial distribution patterns (Farr et al., 2007). The phenology information for this study was sourced from the RiceAtlas rice calendar and yield database (Laborte et al., 2017). RiceAtlas records data on rice planting and harvest dates by growing season and yield estimates for all rice-producing countries. It contains detailed information on rice phenology for all seasons in the provincial regions of EA countries. These phenological metrics include the start, peak, and end dates of sowing,  transplanting and harvesting for each rice-growing
season.

To assess the accuracy of the generated rice map, an independent validation sample set was constructed through field surveys and visual interpretation, which contains 91,320 validation samples in total (46,908 rice and 44,412 non-rice), as shown in Figure 2. Besides, both official statistical yearbook data as well as existing rice data products were collected. Annual rice area statistics were collected from national statistical yearbooks at the city level in China and at the provincial
level in Japan, the Democratic People's Republic of Korea, and the Republic of Korea. All area values were converted to ha. Five existing publicly available datasets were acquired, including APRA500 (Han et al., 2022), NESEA-Rice10 (Han et al., 2021), Rice-TWDTW (Pan et al., 2021; Shen et al., 2023), RUNetRice-SouthKorea (Jo et al., 2023), and HistoricalRice-Japan (Carrasco et al., 2022). Han et al. used MODIS data to map 500 m rice distribution in the Asian monsoon region, covering all EA countries (referred to as APRA500); Han et al. proposed a phenology-based algorithm with MODIS and S1
to achieve 10 m resolution rice distribution in Northeast and Southeast Asia, covering Northeast China, the Democratic People's Republic of Korea, Japan, and the Republic of Korea (referred to as NESEA-Rice10); Pan et al. and Shen et al. adopted the TWDTW algorithm with Sentinel-1/2 data to create a 10 m resolution single- and double-season rice distribution map, covering the main rice production areas in China (referred to as Rice-TWDTW); Jo et al. designed a deep learning model integrating UNet and RNN with S1 data to acquired 10 m rice distributions in the Republic of Korea from 2017 to
2021 (referred to as RUNetRice-SouthKorea); Carrasco et al. used a phenology-based algorithm to map rice distribution in Japan with Landsat imagery in 30 m resolution (referred to as HistoricalRice-Japan).



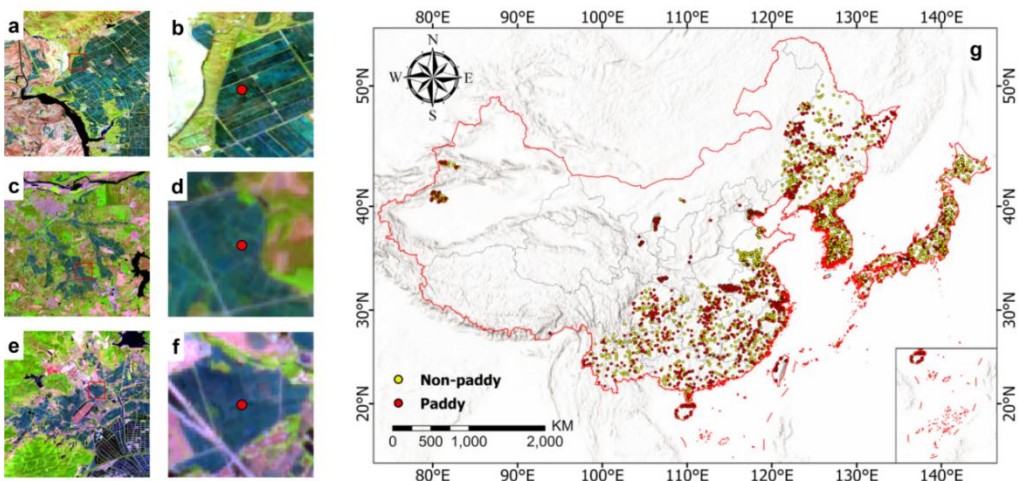

**Figure 2.The Distribution of the validation sample set: (a, c, e) Sentinel-2 false-color images (R: SWIR1, G: NIR, B: Red); (b, d, f) enlarged local views of (a, c, e) respectively; (g) the distribution of rice and non-rice validation sample set. Basemap sources for (g): Esri, TomTom, Garmin, FAO, NOAA, USGS, © OpenStreetMap contributors, and the GIS User Community.**

**Table 2. Auxiliary data information used in the study.**

| Data type | Data product or country name | Coverage area | Year | Resolution | Description of use | Data access |
|---|---|---|---|---|---|---|
| Land Cover Product | WorldCover 2020 | Global | 2020 | 10 m | Crop map generation | https://developers.google.com/earth-engine/datasets/catalog/ESA_WorldCover_v100 |
| | WorldCover 2021 | Global | 2021 | 10 m | Crop map generation | https://developers.google.com/earth-engine/datasets/catalog/ESA_WorldCover_v200 |
| DEM | SRTM V3 | Land between 60° N and 56° S latitude | 2020 | 30 m | S1 pre-processing and slope map generation | https://developers.google.com/earth-engine/datasets/catalog/USGS_SRTMGL1_003 |
| Rice Phenology Information | RiceAtlas | 115 rice growing countries | 2017 | Province scale | Rice phenology confirmation | https://dataverse.harvard.edu/dataset.xhtml?persistentId=doi:10.7910/DVN/JE6R2R |



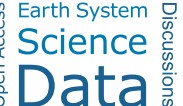

| | | | | | |
|---|---|---|---|---|---|
| | China | China | 2022 | City Scale | Accuracy evaluation | https://data.stats.gov.cn/ |
| Statistical Yearbook Data | The Democratic People's Republic of Korea and the Republic of Korea | The Democratic People's Republic of Korea and the Republic of Korea | 2023 | Province scale | Accuracy evaluation | https://kosis.kr/statisticsList/statisticsListIndex.do?vwcd=MT_ZTITLE&menuId=M_01_01 |
| | Japan | Japan | 2021 | Province scale | Accuracy evaluation | https://www.stat.go.jp/english/ |
| Existing Rice Distribution Maps | APRA500 | Monsoon Asia | 2000-2020 | 500 m | Spatial consistency assessment | https://zenodo.org/records/5555721 |
| | NESEA-10 | Northeast and Southeast Asia | 2017-2019 | 10 m | Spatial consistency assessment | https://zenodo.org/records/5645344 |
| | Rice-TWDTW | China (Single-season rice: 21 provinces; Double-season rice:9 provinces) | Single-season rice: 2017-2022; Double-season rice: 2016-2020 | 10 m | Spatial consistency assessment | http://www.nesdc.org.cn/sdo/detail?id=6195c2f07e2817528307c465 https://www.scidb.cn/en/detail?dataSetId=b07f90ea5f0c4e359fa4119a0030f9da |
| | RUNetRice-SouthKorea | The Republic of Korea | 2017-2021 | 10 m | Spatial consistency assessment | https://zenodo.org/records/5845896 |
| | HistoricalRice-Japan | Japan | 1985-2019 | 30 m | Spatial consistency assessment | https://data.mendeley.com/datasets/v4xmd5kgck/1 |

## 3 Method

To generate the annual rice distribution map for EA in 2023, this study designed a high-precision general rice mapping framework, as illustrated in Figure 3. This framework comprised two key components: (1) an indices-based sample set generation method, and (2) an optical-SAR adaptive fusion model.

**Figure 3. The overall workflow of the proposed rice distribution mapping framework.**



## 3.1 Indices-based sample set generation method

Considering the various cultivation conditions and phenological patterns of EA, identifying rice candidate areas by relying
solely on a single data source might result in misclassification or omission. In this study, we designed a robust sample set
generation method that employs indices from both SAR and optical data to identify rice candidate areas and combines the
results to refine the sample set.

### 3.1.1 SAR-based rice candidate area extraction

Existing studies have demonstrated that SAR VH polarization is more effective in capturing the unique characteristics
of rice growth compared to VV polarization (Zhan et al., 2021; Xu et al., 2023). However, current phenology-based
methods for rice mapping using SAR often rely on multiple rice phenological stages, making large-scale application
challenging. Therefore, we proposed SRMI, a new rice index based on the temporal characteristics of the entire rice
phenological period of the time-series S1 VH data.

As illustrated in our previous work (Xu et al., 2021; Sun et al., 2023), the minimum backscatter coefficient ($\sigma_{min}^0$) identifies
the flooding stage, distinguishing rice from other crops. The maximum backscatter coefficient ($\sigma_{max}^0$) reduces
misclassification due to water bodies. The mean backscatter coefficient ($\sigma_{mean}^0$) helps differentiate rice from other objects
such as water bodies and buildings. The variance of the backscatter coefficient ($\sigma_{var}^0$) indicates the variability of the
backscatter coefficient over time, distinguishing farmland from other land cover types. Thus, four temporal statistical
parameters ($\sigma_{min}^0$, $\sigma_{max}^0$, $\sigma_{mean}^0$, and $\sigma_{var}^0$) that effectively distinguish rice from other land cover types during the
phenological period were calculated with the following equations.

$$\sigma_{min}^0 = min\{\sigma_1^0, \sigma_2^0, \sigma_3^0, \ldots, \sigma_n^0\} \tag{1}$$

$$\sigma_{max}^0 = max\{\sigma_1^0, \sigma_2^0, \sigma_3^0, \ldots, \sigma_n^0\} \tag{2}$$

$$\sigma_{mean}^0 = \frac{1}{n}\sum_1^n \sigma_i^0 \tag{3}$$

$$\sigma_{var}^0 = \frac{1}{n}\sum_1^n \left(\sigma_i^0 - \sigma_{mean}^0\right) \tag{4}$$

where $\sigma_n^0$ is the backscatter coefficient at the *n-th* observation.

Next, the four features ($\sigma_{min}^0$, $\sigma_{max}^0$, $\sigma_{mean}^0$, and $\sigma_{var}^0$) were normalized to eliminate outlier values according to Equation (5):

$$F(x) = \begin{cases} 1, & x > B \\ \dfrac{x - A}{B - A}, & A \leq x \leq B \\ 0, & x < A \end{cases} \tag{5}$$

where $x$ represents the statistical parameters; $A$ and $B$ are preset parameters for normalization. Based on experience and
extensive comparative experiments, we set the thresholds of $A = -25$ and $B = -10$ for $\sigma_{min}^0$ and $\sigma_{max}^0$ and $\sigma_{mean}^0$; $A = -$




20 and $B = -10$ for $\sigma^0_{mean}$; $A = 0$ and $B = 10$ for $\sigma^0_{var}$. Through Equation (5), the normalized values of the four features ($F(\sigma^0_{min})$, $F(\sigma^0_{max})$, $F(\sigma^0_{mean})$, $F(\sigma^0_{var})$) were obtained.

Subsequently, to mitigate the impact of SAR data speckle noise, the Simple Non-Iterative Clustering (SNIC) (Achanta and Susstrunk, 2017) superpixel segmentation was applied to divide the plots into different objects. The input features for SNIC included the normalized values of the four features ($F(\sigma^0_{min})$, $F(\sigma^0_{max})$, $F(\sigma^0_{mean})$, $F(\sigma^0_{var})$). In this study, the SNIC

algorithm was configured with a size parameter of 15, a compactness value of 0.8, and a connectivity of 8. Consequently, the mean feature values within different objects ($F_{snic}(\sigma^0_{min})$, $F_{snic}(\sigma^0_{max})$, $F_{snic}(\sigma^0_{mean})$, $F_{snic}(\sigma^0_{var})$) were obtained.

Then, the object-based SRMI was defined as follow:

$$SRMI = \left(1 - F_{snic}(\sigma^0_{min})\right) \times \left(1 - F_{snic}(\sigma^0_{mean})\right) \times F_{snic}(\sigma^0_{max}) \times F_{snic}(\sigma^0_{var}) \tag{6}$$

Finally, the SAR-based rice candidate area $Rice_{SAR}$ was established using SRMI:

$$Rice_{SAR} = \begin{cases} 1, & SRMI \geq \beta \\ 0, & Otherwise \end{cases} \tag{7}$$

where $\beta$ is the SRMI threshold for extract candidate rice areas.

The range of SRMI index values for common land cover types, as shown in Figure 4, demonstrates that setting the threshold $\beta$ to 0.5 can effectively distinguishes rice from non-rice fields. Figure 5 displays some examples of potential rice candidate areas identified by SRMI. Located in different countries and with various cultivation backgrounds, these patches maintained good consistencies with manual interpretations, demonstrating the effectiveness of SRMI.

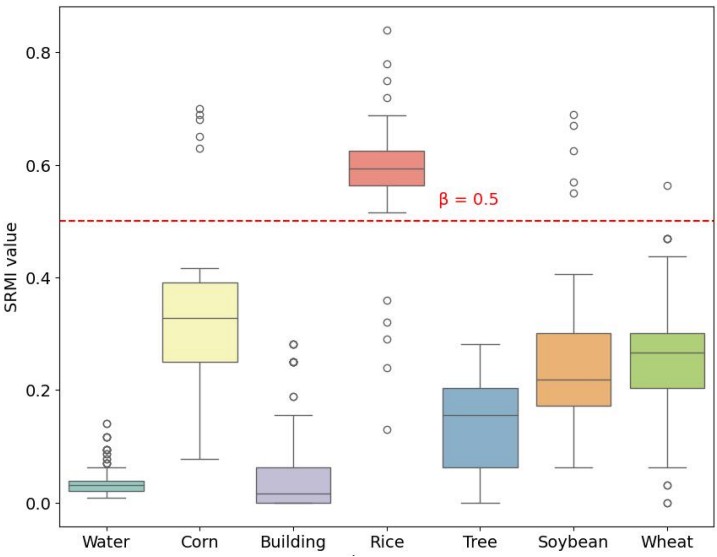

**Figure 4. The box plot of the SRMI value of land cover types.**

Earth System Open Access
Science Discussions
Data

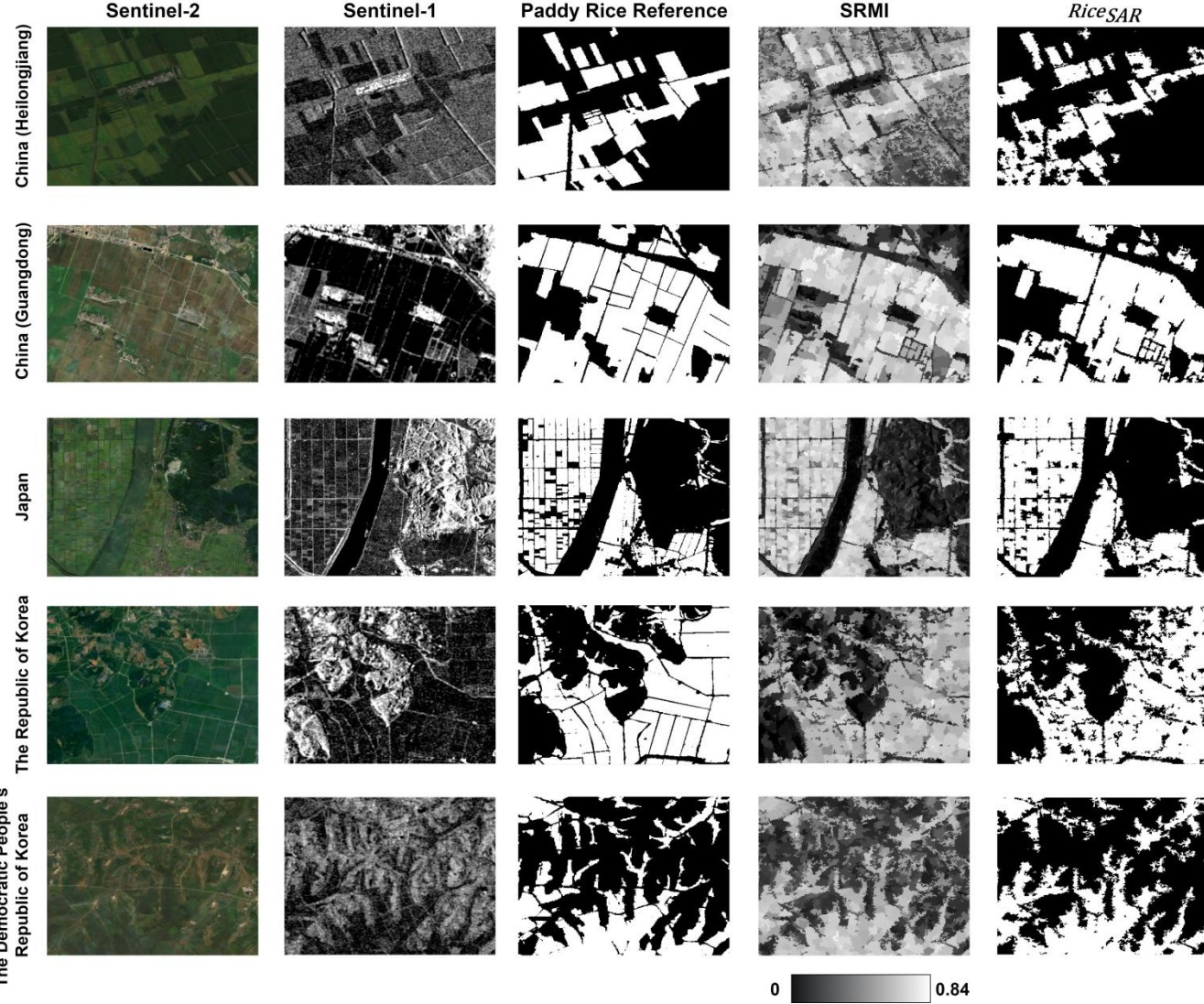

**Figure 5. Demonstrations of rice candidate areas in different regions based on SRMI. The first column shows Sentinel-2 images, and the second column shows Sentinel-1 images.**

### 3.1.2 Optical-based rice candidate area extraction

During the seeding and transplanting period of rice, the paddy field has a unique flood inundation period. The flood signal of the paddy field can be determined by the relationship between LSWI and EVI to determine the optical-based rice candidate area ($Rice_{Opitcal}$), as shown in Figure 6. $Rice_{Opitcal}$ was defined as:

$$Rice_{Opitcal} = \begin{cases} 1, & LSWI + \alpha \geq EVI \\ 0, & Otherwise \end{cases} \tag{8}$$

where $\alpha$ is the threshold set to 0.05 (Xiao et al., 2005a). The entire seeding and transplanting period for rice spans approximately 30 to 50 days (approximately 6-10 S2 scenes). Due to the variations in the rice planting cycle and cloud



interference, the number of available images might not be guaranteed for each pixel. To maximize the number of usable rice samples in cloudy regions, if a pixel satisfied equation (8) in at least one image, it was initially marked as a potential rice area.

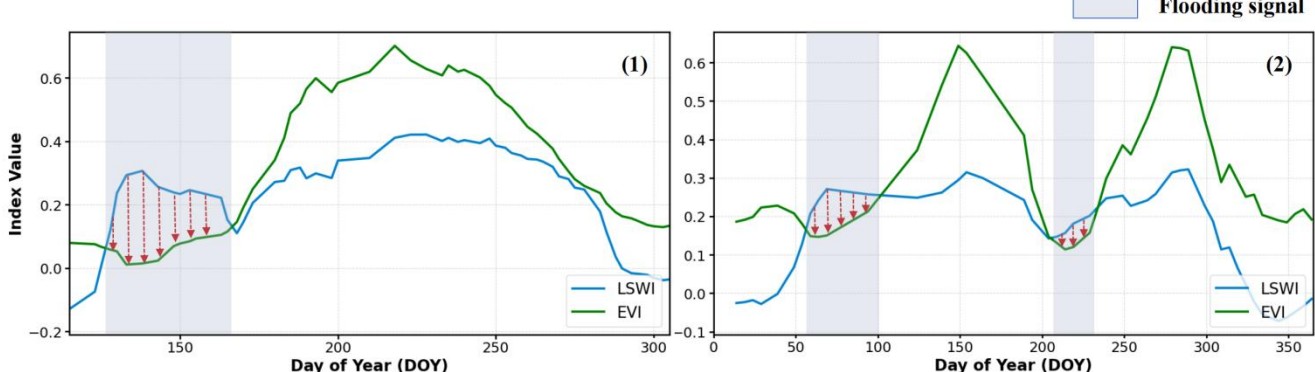

Figure 6. Temporal variations of EVI and LSWI for single-season and double-season rice based on field sample points: (1) single-season rice; (2) double-season rice.

### 3.1.3 Sample selection based on combined candidate area

To improve the accuracy of rice samples, the $Rice_{Opitcal}$ and $Rice_{SAR}$ candidate areas were intersected to obtain the comprehensive rice/non-rice candidate area ($Rice_{Both}$), which was defined as:

$$Rice_{Both} = \begin{cases} 1, & if\ Rice_{Opitcal} = 1\ and\ Rice_{SAR} = 1 \\ 0, & if\ Rice_{Opitcal} = 0\ and\ Rice_{SAR} = 0 \\ none, & Otherwise \end{cases} \tag{9}$$

Finally, considering spatial heterogeneity, a fishnet covering the study area with 1-degree intervals in longitude and latitude was created with GEEMAP (Wu, 2020). Within each grid, 2000 sample points were selected from $Rice_{Both}$ using stratified random sampling, maintaining a 1:1 ratio of rice to non-rice sample points.

### 3.2 Optical-SAR adaptive fusion model

To fully leverage the advantages of SAR and optical data in the rice classification task, an optical-SAR adaptive fusion model was designed in this study by stacking multiple RF classifiers, as shown in Figure 7.

RF model integrates multiple decision trees to reduce the bias and variance of individual trees, thus providing more accurate classification performance. RF is robust to noise, less prone to overfitting, and highly generalizable and transferable, which has been proven effective by previous remote sensing rice mapping researchs (He et al., 2021; Abdali et al., 2023; Zhang et al., 2023a). In contrast to the commonly utilized strategy that directly feeding original SAR and optical features into the same RF model for classification, we proposed a hierarchical integration strategy. First, a parallel structure was designed to exploit the time-series features of S1 and S2 data with independent RF models. Afterwards, another RF model was trained to



adaptively fuse the classification probabilities of both data sources. In this process, the data availability of S2 was considered by introducing the cloud-free indicator as the input feature.

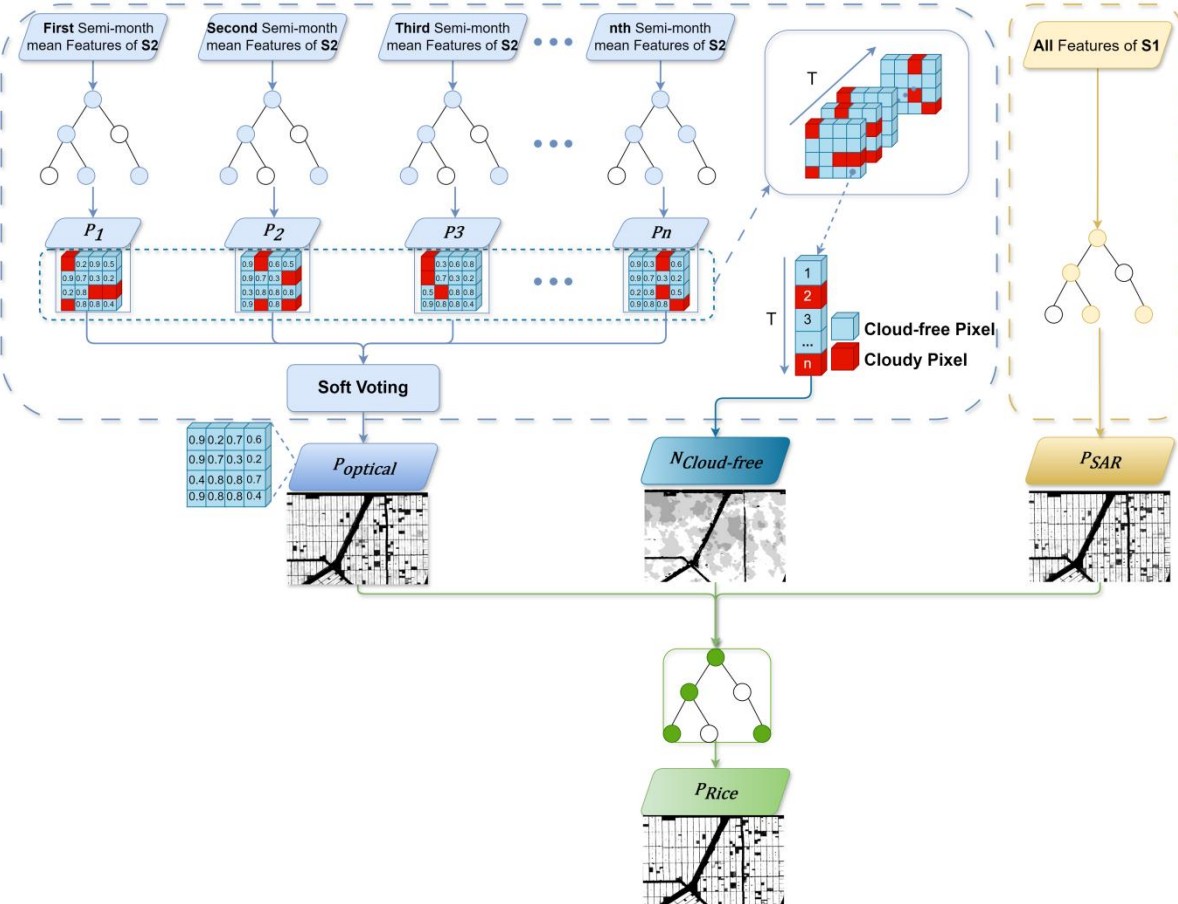

**Figure 7. Structure of the optical-SAR adaptive fusion model using optical and SAR data.**

Given that S2 data has a revisit period of 5 days but is affected by clouds, a separate RF model was trained for the mean features of each semi-monthly period during the entire rice phenological period. The classification results of each model were output as probability values ( $P_i$ ). The mean of all RFs classification results generated the optical-based rice classification probability ($P_{Optical}$):

$$P_{Optical} = \frac{1}{N_{Cloud-free}} \sum_{i=1}^{N} P_i \tag{10}$$

where $N_{Cloud-free}$ represents the number of cloud-free pixels in all semi-monthly images during the rice phenological period, $N$ is the total number of semi-monthly images during the rice phenological period. For instance, for a 150-day rice growth cycle, $N=10$, $N_{Cloud-free} \leq N$, $i=1, 2, ..., N$. In general, the closer the $N_{Cloud-free}$ value is to $N$, the more reliable is the $P_{Optical}$ result. Conversely, the greater the distance between $N_{Cloud-free}$ and $N$, the higher the uncertainty of $P_{Optical}$.



For S1 data, the RF model was trained with VH, VV, and CR as inputs, where the overlapping areas adopt the 12-day
average of all S1 images. The RF classification result was represented as the SAR-based rice classification probability
$(P_{SAR})$.

With all the independent RF models trained by both optical and SAR data, the probabilities were further combined through
another RF model, which intended to dig the hidden relationship between the rice recognition results derived by two data
sources. The cloud-free frequency was taken as the input feature as well, which could modulate the $P_{Optical}$ with optical data
availability to generate the final rice classification result $(P_{Rice})$.

In this study, the *ntrees* parameter of the RF was set to 100, and other parameters were set to default values.

**3.3 Accuracy evaluation**

To evaluate the reliability of the EARice10 rice distribution map for the EA region, we used overall accuracy (OA), user's
accuracy (UA), and producer's accuracy (PA) to assess the accuracy of EARice10. These metrics are calculated based on the
confusion matrix, using the following formulas:

$$OA = \frac{TP + TN}{TP + TN + FP + FN} \tag{11}$$

$$UA = \frac{TP}{TP + FN} \tag{12}$$

$$PA = \frac{TP}{TP + FP} \tag{13}$$

Where TP is the number of pixels correctly classified as rice, FP is the number of non-rice pixels misclassified as rice, TN is
the number of pixels correctly classified as non-rice, and FN is the number of rice pixels misclassified as non-rice.

Additionally, we used the coefficient of determination ($R^2$) to evaluate the correlation between the rice area of EARice10 and
the government statistics or existing rice distribution maps. The formula for $R^2$ is as follows:

$$R^2 = \frac{\left(\sum_{i=1}^{n} (x_i - \overline{x}_i) \times (k_i - \overline{k}_i)\right)^2}{\sum_{i=1}^{n} (x_i - \overline{x}_i)^2 \times \sum_{i=1}^{n} (k_i - \overline{k}_i)^2} \tag{14}$$

where $n$ is the total number of administrative units, $x_i$ represents the rice area of EARice10 in the administrative unit, $\overline{x}_i$ is
the average rice area of EARice10 across all administrative units, $k_i$ is the rice area from government statistics or existing
products in the administrative unit, and $\overline{k}_i$ is the average rice area from government statistics or existing products across all
administrative units.



## 4 Results

### 280    4.1 2023 East Asia 10 m Resolution rice distribution map

The 2023 10 m resolution rice distribution map for EA, referred to as EARice10, is illustrated in Figure 8. Statistical information on its distribution is analyzed in conjunction with DEM and other data, as shown in Figure 9. And the rice distribution maps of each country are shown in Figures 10-13, respectively.

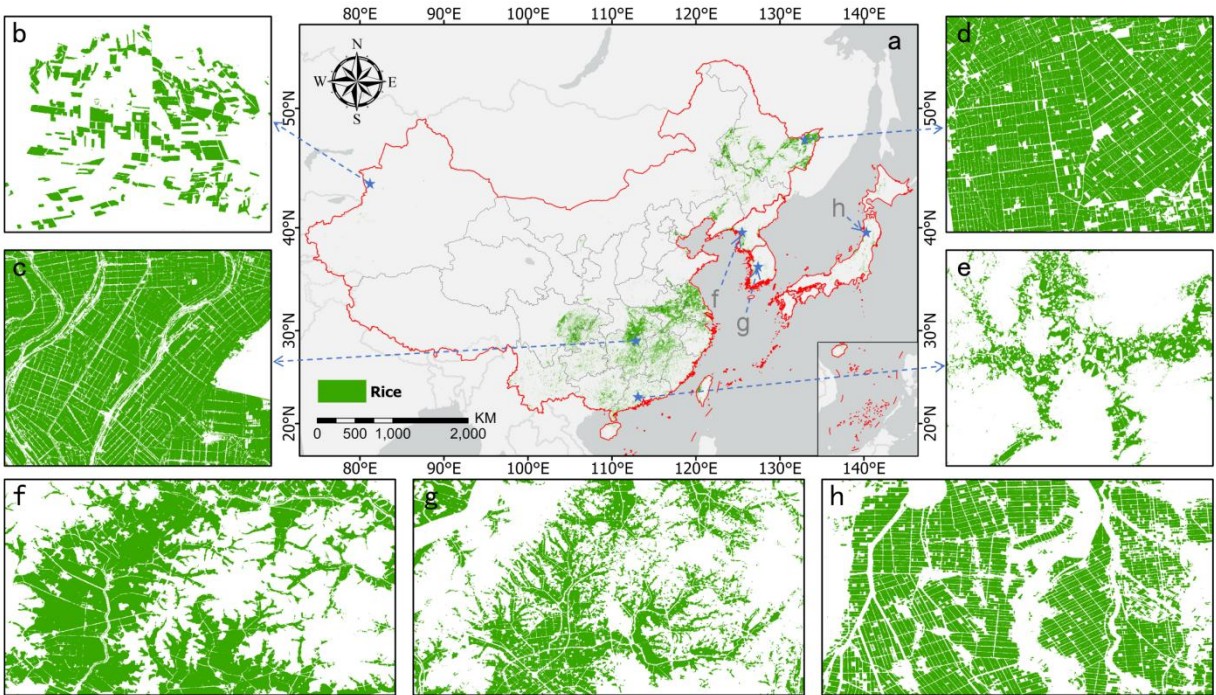

**Figure 8. 2023 East Asia 10 m resolution rice distribution map (EARice10): (a) full coverage of EARice10; (b)-(h) zoomed views of rice distribution in selected regions: (b) Xinjiang, China (provincial rice planting area less than 100,000 ha); (c) Heilongjiang, China (single-season rice region); (d) Hunan, China (mixed-season rice region); (e) Guangdong, China (double-season rice region); (f) the Democratic People's Republic of Korea (single-season rice region); (g) the Republic of Korea (single-season rice region); (h) Japan (single-season rice region).**

As indicated by Figures 8 and 9, rice cultivation in EA was mainly distributed between 98°E-142°E and 20°N-50°N, with the highest density of rice cultivation near 112.58°E and 34.18°N. Figure 9-b showed a bimodal distribution pattern in latitude, with the peaks corresponding to central China and northeast Asia, which have hydrologic and soil conditions suitable for rice growth. Figure 9-c showed that rice cultivation areas were mainly concentrated at low altitudes, with about 95.5% of the rice cultivated in areas below 1,000 m, 3.8% cultivated between 1,000 m and 2,000 m, and only 0.7% of the

rice cultivated in areas above 2,000 m. Most of the rice in EA was grown in low-slope areas, as shown in Figure 9-d, because these areas are more suitable for irrigation and management. Specifically, about 87.8% of it was grown in areas with slopes of less than 5 degrees. About 12.2% of the rice-growing areas have slopes greater than 5 degrees, mainly in the hilly areas of southern China, where single-season rice is common (He et al., 2021).





**Figure 9. Statistical analysis of rice area in different geographical regions: (a) longitude; (b) latitude; (c) DEM; (d) slope.**

In 2023, China's annual rice cultivation area was 24,716 thousand ha. Specifically, single-season rice was mainly concentrated in the northeastern plains, including the Sanjiang Plain, Songnen Plain, and Liaohe Plain, accounting for approximately 20% of China's total rice cultivation area. Mixed-season rice was primarily cultivated in the lake-rich regions of the Yangtze River Basin, such as Dongting Lake in Hunan Province, Taihu Lake in Jiangsu Province, and Poyang Lake in Jiangxi Province. Double-season rice was primarily cultivated in the Pearl River Delta, where the temperate climate conditions are favorable. Additionally, substantial rice cultivation areas were mainly located in southwestern China, particularly in eastern Sichuan Province, Guizhou Province, and Yunnan Province. The top three provinces with largest annual rice cultivation area were Heilongjiang (14.54%), Hunan (10.98%), and Anhui (9.46%). This rice distribution map aligns well with the spatial distribution patterns identified in previous studies (Xiao et al., 2005a; Dong et al., 2016; Carrasco et al., 2022; Wei et al., 2022). Furthermore, EARice10 includes provinces in China where the area under rice cultivation is less than 100,000 ha, which were not considered in other datasets (Pan et al., 2021; Shen et al., 2023).



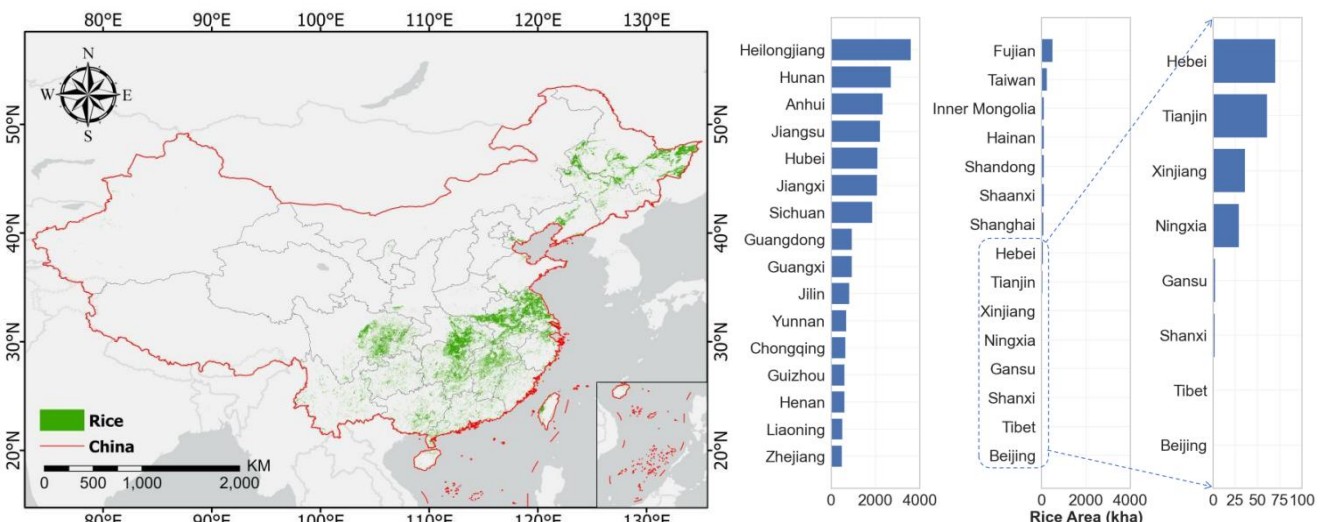

**Figure 10. Ten-meter rice distribution map in China and provincial rice area statistics (2023).**

In Japan, the annual rice cultivation area was 1,251,464 ha. Specifically, rice cultivation was primarily concentrated on
coastal alluvial plains, with the most extensive areas found in western Hokkaido, the northwest and central coastal regions of
Honshu, the Kanto Plain, areas surrounding Lake Biwa in central Honshu, and western Kyushu Island. Significant rice
cultivation regions included Tohoku (29.76%), Chubu (20.97%), Kanto (16.83%), and Hokkaido (8.08%), and the top three
prefectures for rice cultivation in 2023 were Niigata (9.17%), Hokkaido (8.08%), and Akita (6.48%).

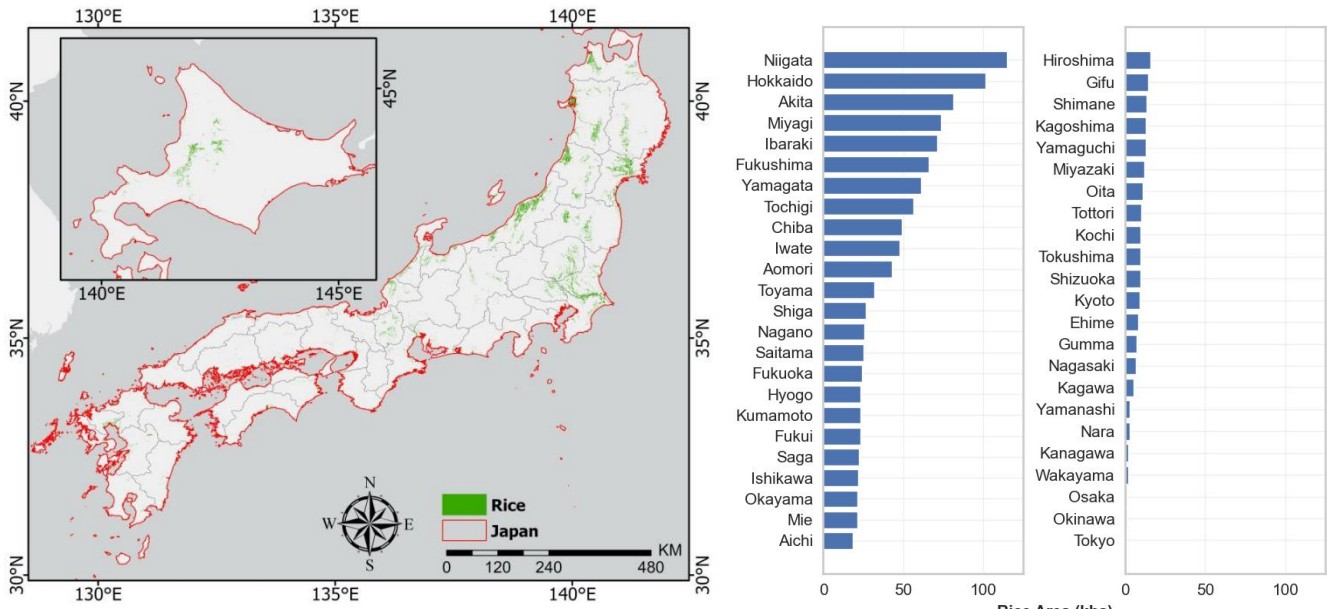

**Figure 11. Ten-meter rice distribution map in Japan and provincial rice area statistics (2023).**





On the Korean Peninsula, rice cultivation was primarily concentrated on the western and southern coastal plains, as these regions offer favorable conditions for large-scale rice cultivation due to the abundance of plains, rivers, and reservoirs, which provide ample irrigation. Some rice paddies are also found in the eastern mountainous areas.

In the Republic of Korea, the annual rice cultivation area was 626,830 ha. Specifically, rice cultivation was mainly
concentrated in the provinces of Jeollanam-do, Chungcheongnam-do, Jeollabuk-do, Gyeongsangbuk-do, Gyeonggi-do, and Gyeongsangnam-do, which together represent 88% of the total rice cultivation area in the Republic of Korea. The top three provinces for rice cultivation in 2023 were Jeollanam-do (21.62%), Chungcheongnam-do (19.47%), and Jeollabuk-do (15.90%).

In the Democratic People's Republic of Korea, the annual rice cultivation area was 504,692 ha. Specifically, rice cultivation
was predominantly concentrated in the western provinces, specifically Hwanghae-namdo, Pyongan-bukdo, Pyongan-namdo, Hamgyong-namdo, and Hwanghae-bukdo. These regions account for 82.49% of the nation's total rice cultivation area. The top three provinces for rice cultivation in 2023 were Hwanghae-namdo (25.35%), Pyongan-bukdo (20.33%), and Pyongan-namdo (18.27%).

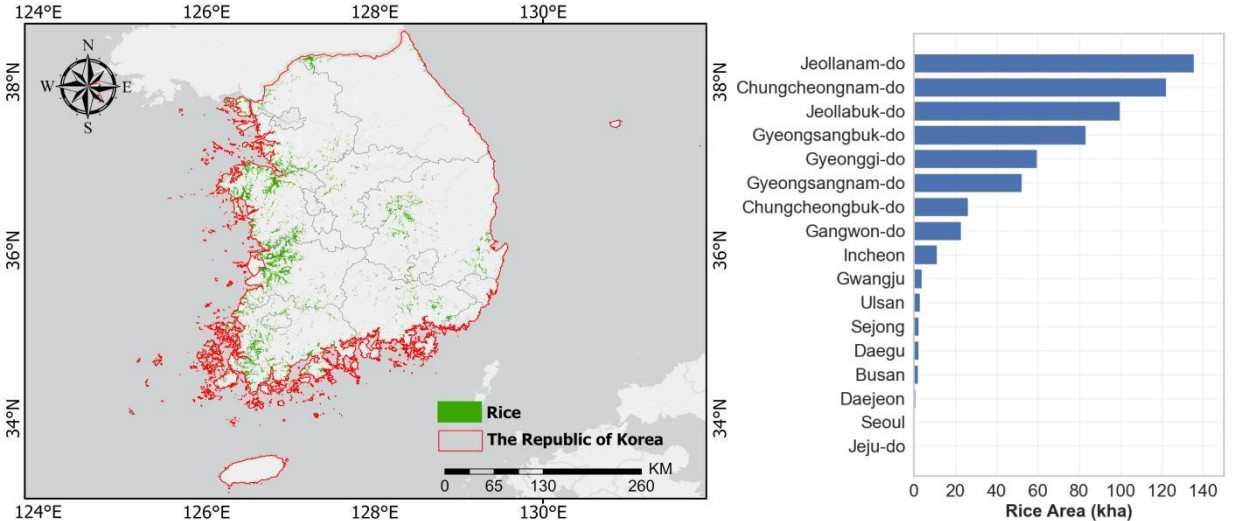

**Figure 12. Ten-meter rice distribution map in the Republic of Korea and provincial rice area statistics (2023).**



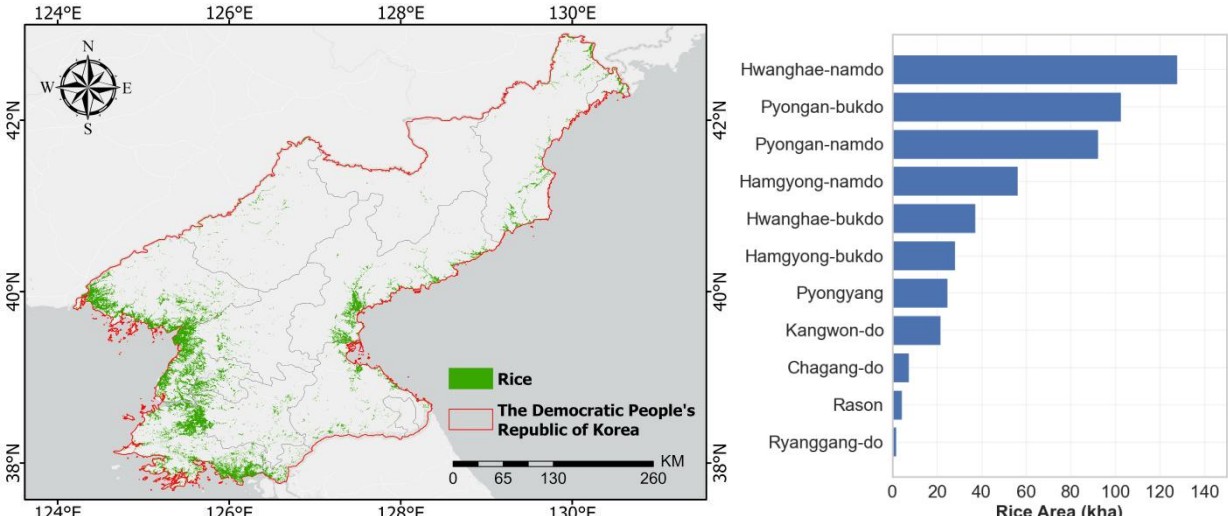

**Figure 13. Ten-meter rice distribution map in the Democratic People's Republic of Korea and provincial rice area statistics (2023).**

### 4.2 Accuracy evaluation results of EARice10

To verify the reliability of EARice10, its accuracy was evaluated using the validation sample set and statistical yearbook data.

As indicated in Table 3, EARice10 achieved the OA of 90.48%, UA of 90.93% and PA of 90.49%, indicating high consistency between the extracted rice areas and the validation sample set. Specifically, the Democratic People's Republic of Korea and the Republic of Korea exhibited the highest classification accuracy, with OAs of 95.33% and 95.03%, respectively. Japan showed an OA of 90.02%, while China's OA was slightly lower at 89.16%. This discrepancy was likely attributable to its larger geographical area and diverse rice cropping systems.

As shown in Figure 14, the accuracy assessment results at the provincial scale indicated that most provinces in the Democratic People's Republic of Korea, the Republic of Korea, and northern provinces of China (such as Heilongjiang, Jilin, and Liaoning) performed excellently, with OAs above 92%. These rice areas belonged to the single-season region and were characterized by vast plains and abundant water resources, making them highly suitable for large-scale rice cultivation, which was consistent with previous studies (Dong et al., 2016; Ni et al., 2021; Zhang et al., 2022a; Zhang et al., 2023a). Conversely, the classification accuracy of rice paddies in southwestern China (such as Tibet and Chongqing) exhibited marginally lower performance due to the complex topography and fragmented distribution of land fields. Nonetheless, it is noteworthy that the OAs in these areas all exceeded 74%.

**Table 3.National-level confusion matrix for the EARice10 against validation sample set**

| Country | Class | Where **Rice*** and **Non-rice*** represent the validation sample points. | | | | |
| | | Rice* | Non-rice* | UA (%) | PA (%) | OA (%) |
| China | Rice | 26736 | 2979 | 89.97% | 89.15% | 89.16% |
| | Non-rice | 3255 | 24516 | 88.28% | 89.17% | |





| | | | | | | |
|---|---|---|---|---|---|---|
| Japan | Rice | 7152 | 816 | 89.76% | 90.34% | 90.02% |
| | Non-rice | 765 | 7101 | 90.27% | 89.69% | |
| The Republic of Korea | Rice | 4269 | 216 | 95.18% | 94.87% | 95.03% |
| | Non-rice | 231 | 4284 | 94.88% | 95.20% | |
| The Democratic People's Republic of Korea | Rice | 4290 | 222 | 95.70% | 94.93% | 95.33% |
| | Non-rice | 210 | 4278 | 94.97% | 95.73% | |
| East Asia | Rice | 42447 | 4233 | 90.93% | 90.49% | 90.48% |
| | Non-rice | 4461 | 40179 | 90.01% | 90.47% | |

**Figure 14. Provincial-level confusion matrix metrics for the EARice10 based on validation sample set**

In addition, we compared the rice area derived from EARice10 with rice area reported in official statistical yearbooks, using municipal-level statistical yearbook data for China and provincial-level statistical yearbook data for Japan, the Republic of Korea, and the Democratic People's Republic of Korea. As illustrated in Figure 15, the $R^2$ of EARice10 at the municipal level in China was 0.94. For the Democratic People's Republic of Korea, the Republic of Korea, and Japan, the $R^2$ values at the provincial level were 0.97, 0.98, and 0.95, respectively. All correlations were statistically significant ($p < 0.01$). These results demonstrated robust agreement between EARice10-derived rice area and government-reported rice area statistics.

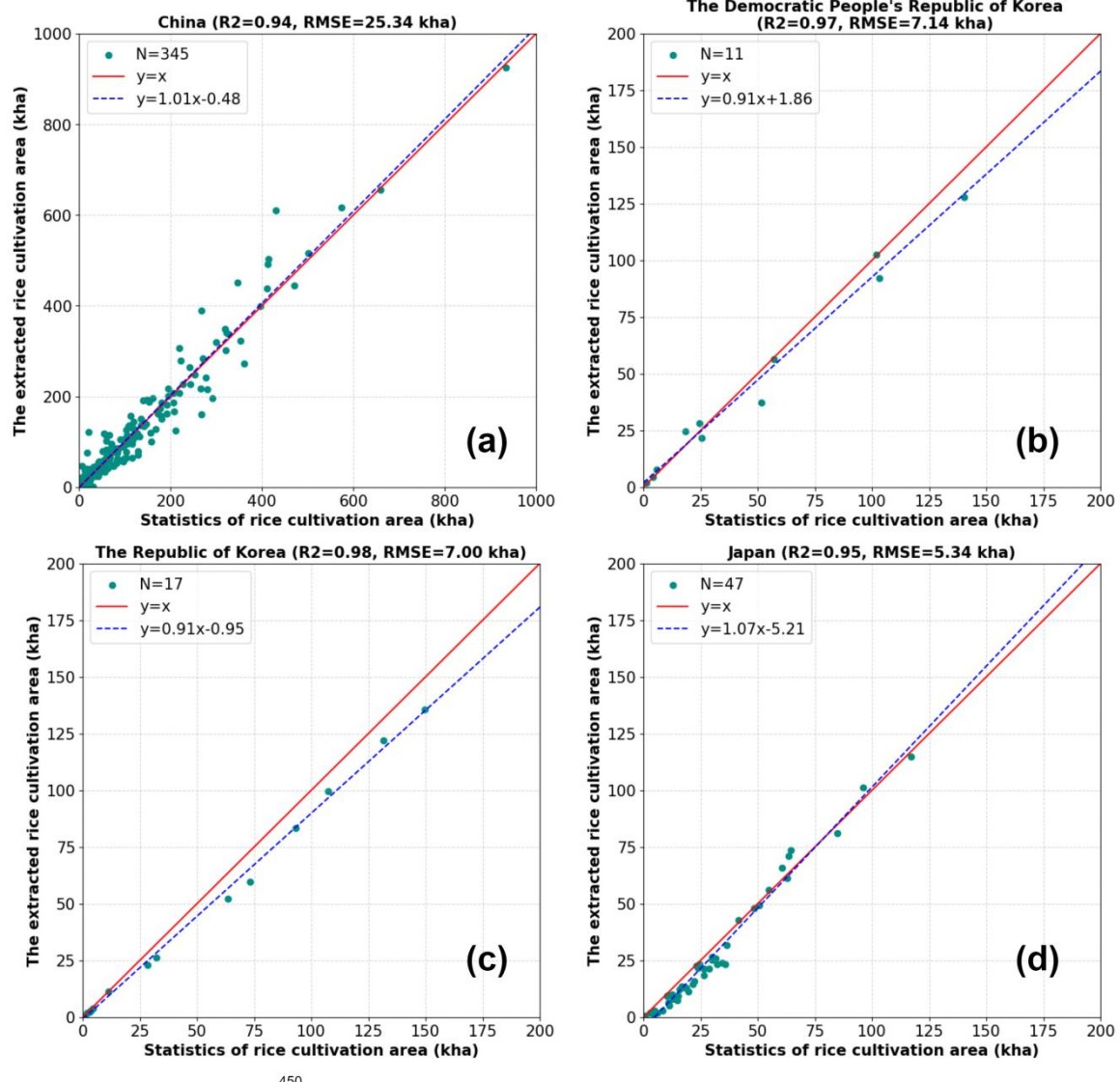

450

**Figure 15. Comparison of the extracted rice area from the EARice10 with the rice area from statistical yearbooks at the**
365 **administrative division scale: (a) municipal-level comparison in China; (b), (c), and (d) provincial-level comparisons in the Democratic People's Republic of Korea, the Republic of Korea, and Japan, respectively.**

## 4.3 Comparison of the EARice10 with existing rice distribution maps

Seven representative sites were selected for comparison with five existing EA rice distribution products, as shown in Figure

16. These sites covered a wide range of countries and cropping patterns, with Sites 1 and 2 in the single-season region of

370 China, Site 3 in the mixed-season region of China, Site 4 in the double-season region of China, and Sites 5-7 in the

Democratic People's Republic of Korea, the Republic of Korea, and Japan, respectively.

The comparison results, as shown in Figure 17, indicated that EARice10 can show more details relative to the 500 m

resolution rice distribution map of APRA500. The EARice10 dataset also showed a high degree of spatial consistency with rice distribution maps from different countries at either 30 m or 10 m resolution. Some localized differences could be found
375   at Sites 1, 2, 5, 6, and 7, where EARice10 depicted more complete paddy plots and clearer road networks reflecting finer spatial details than the existing products. At Sites 3 and 4, EARice10 showed higher spatial completeness and lower noise levels than the existing products. Overall, the comparison results demonstrated the satisfying quality of EARice10.

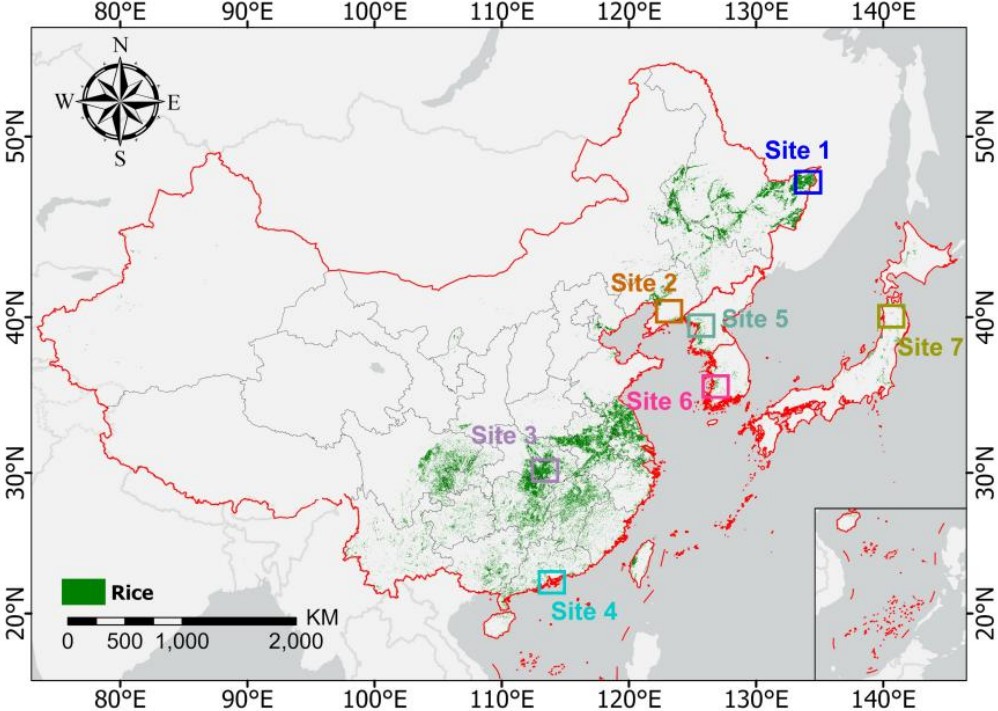

**Figure 16. The sites for comparison.**



**Figure 17. Detailed comparison of the EARice10 with five existing rice distribution maps. Site 1 and Site 2 are single-season rice planting areas in China (Heilongjiang and Liaoning Province), Site 3 is a mixed-season rice planting area in China (Hubei Province), Site 4 is a double-season rice planting area in China (Guangdong Province), and Sites 5-7 are single-season rice planting areas located in the Democratic People's Republic of Korea, the Republic of Korea, and Japan, respectively.**

Figure 18 compared rice area estimates derived from EARice10 and existing rice distribution maps at the administrative unit level for each country. Strong correlations were observed between rice area estimates by EARice10 and existing products. In

China, EARice10 rice area estimates were significantly correlated with those from Rice-TWDTW ($R^2 = 0.91$, p < 0.01) and NESEA-Rice10 ($R^2 = 0.98$, p < 0.01). Similarly, strong correlations were observed in the Democratic People's Republic of Korea with NESEA-Rice10 ($R^2 = 0.91$, p < 0.01), in the Republic of Korea with RUNetRice-SouthKorea ($R^2 = 0.97$, p < 0.01), and in Japan with NESEA-Rice10 ($R^2 = 0.91$, p < 0.01) and HistoricalRice-Japan ($R^2 = 0.79$, p < 0.01). Notably, in Japan, rice area estimates from EARice10 were consistently lower than those from the HistoricalRice-Japan (slope = 0.72). This discrepancy was likely attributable to differences in mapping periods: HistoricalRice-Japan represented a 5-year aggregate (2015–2019), whereas EARice10 reflected the year 2023. Despite the temporal differences in map products, our comparisons with existing products validated the reliability of EARice10 in accurately representing the spatial distribution of rice cultivation in EA for 2023.

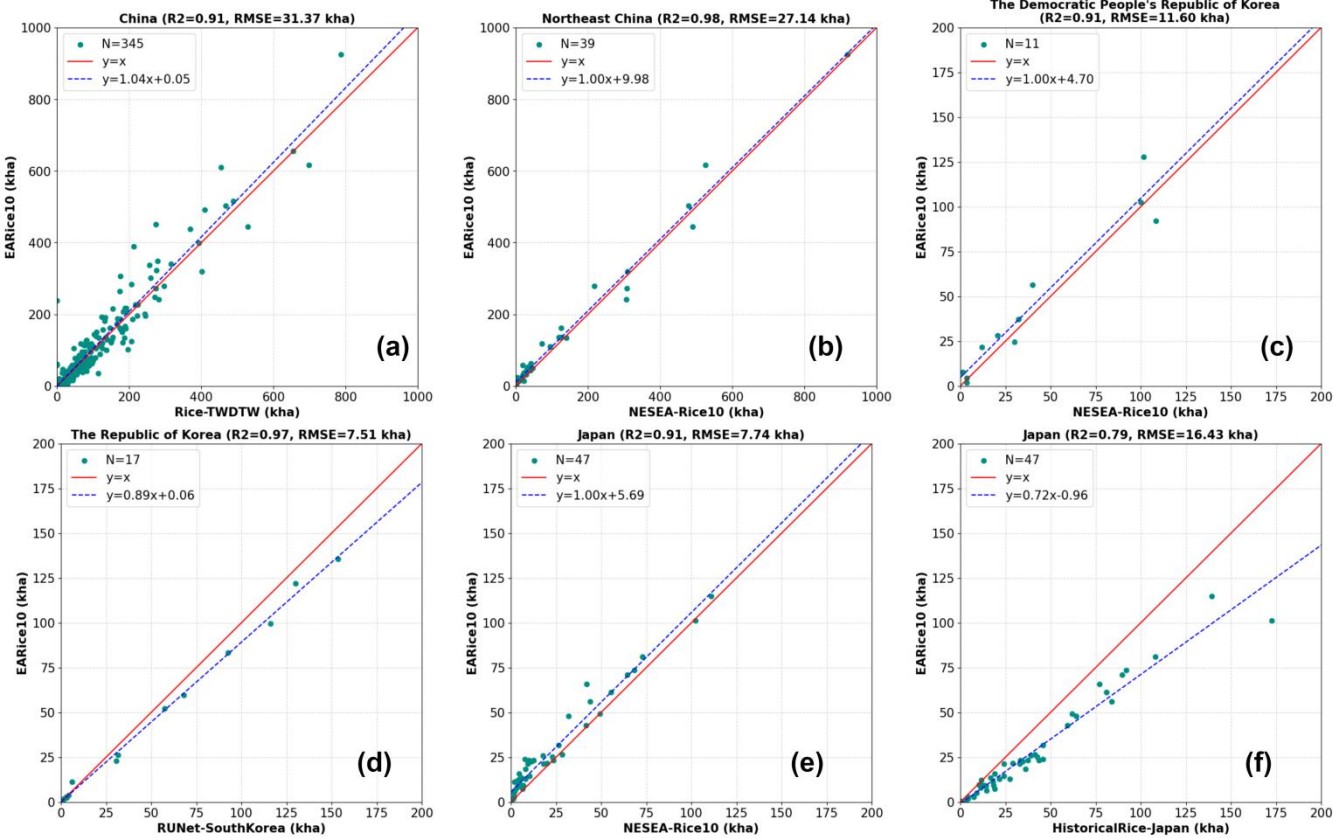

Figure 18. Comparison between EARice10 and existing datasets across different countries' administrative regions.

## 5 Discussion

With its vast expanse, EA is a significant global rice production area that encompasses multiple climate zones. However, cloud cover presents challenges for high-precision rice area mapping in the region. Figure 19 illustrates the number of cloud-free S2 bi-monthly images in different regions of EA from 2020 to 2023, revealing significant differences due to the uneven



temporal and spatial distribution of clouds. Southern China and parts of Japan are particularly affected, with 9.39% and 11.38% of areas in China and Japan, respectively, having less than 15 cloud-free pixels (Figure 20). Therefore, relying solely on optical remote sensing data is insufficient, and a combination of optical and SAR data is required for accurate mapping of rice distribution.

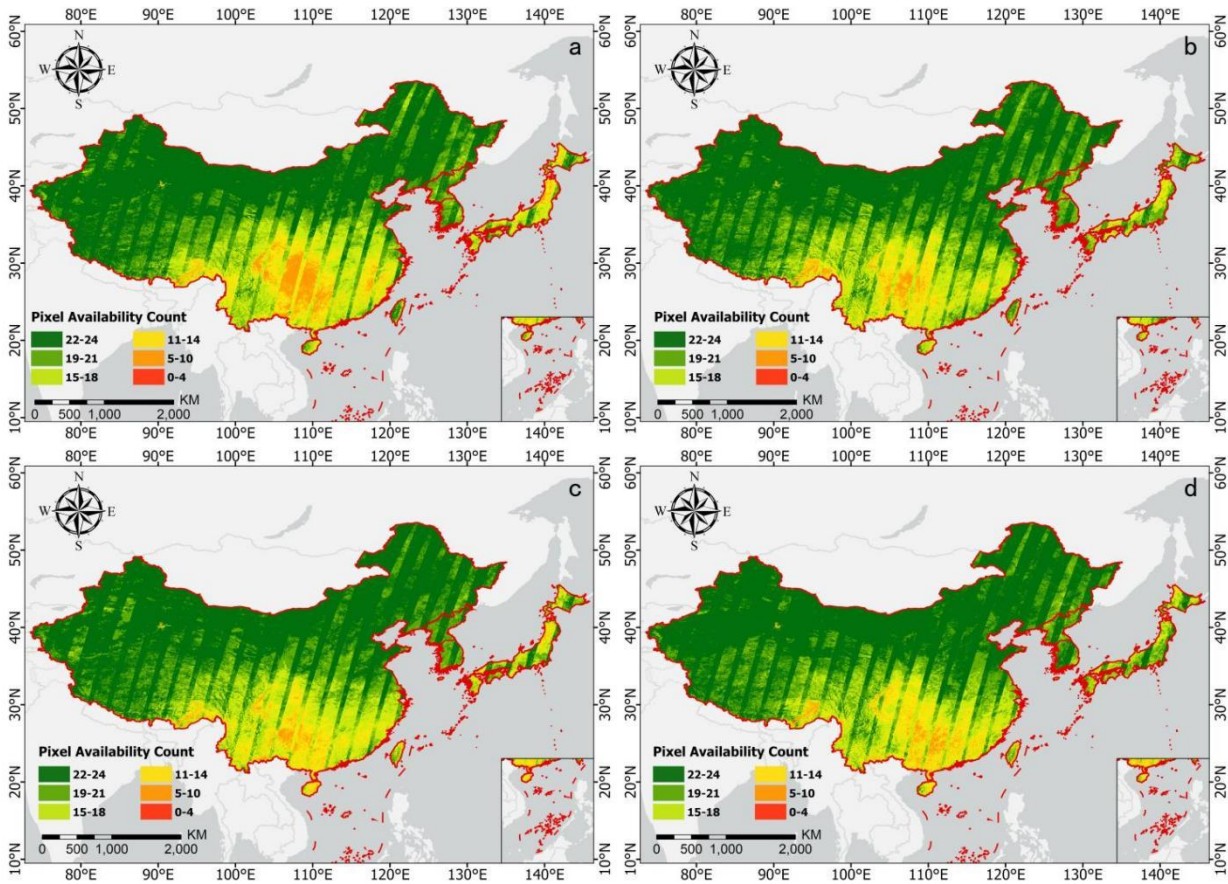

**Figure 19. Number of cloud-free semi-monthly pixels from 2020 to 2023, where (a), (b), (c), and (d) represent 2020, 2021, 2022, and 2023, respectively.**




**Figure 20. Proportion of cloud-free semi-monthly pixel counts from 2020 to 2023 in different countries.**

Using in-season remote sensing data and phenological methods to generate training samples offers a feasible solution for large areas (Gao et al., 2023; Zhang et al., 2023b). However, cloud cover and variations in rice planting cycles in different regions mean optical data alone cannot accurately determine rice distribution. To address this, we propose SRMI, a novel

SAR-based index for rice mapping, which identifies SAR-based rice candidate areas through a single threshold during one rice phenological period. The rice candidate area extracted by SAR is combined with the optical-derived area by the

intersection operation, which intends to purify the rice candidates to increase the quality of samples. This combination avoids misclassification or omission due to the limitations of a single data source, enhancing sample representativeness.

In subcontinental rice mapping, phenological methods alone are insufficient for high-precision rice distribution maps.
Therefore, to achieve high-precision rice mapping and leverage the advantages of both optical and SAR data, this study designs an optical-SAR adaptive fusion model based on a stacking approach. This model utilizes a parallel structure to fully exploit the features of both optical and SAR data, and incorporates the number of cloud-free S2 pixels as a feature in the final decision model, thereby achieving accurate rice classification by considering the uncertainty impact of clouds on rice mapping results.

Using this method, we obtain a 10 m rice distribution map of EA for 2023 (EARice10). The product is comprehensively evaluated using validation sample sets, statistical yearbook data, and existing rice distribution maps. The results indicate that EARice10 is highly consistent with statistical information and existing products, and is able to reflect precise rice distribution information of EA in 2023.

Despite the promising results of EARice10, there is room for improvement. To obtain high-precision rice distribution maps
of EA, the Optical-SAR adaptive fusion model designed in this study is based on a stacking model approach. This method improves classification accuracy by combining the strengths of multiple models, but it sacrifices computational efficiency to some extent compared to single machine learning models. While the high-performance computing capabilities of the GEE platform facilitate the implementation of such computationally intensive models at a subcontinental scale, future research could explore more lightweight and robust classification models to enhance computational efficiency without significantly
compromising classification accuracy. This would enable the method to be more effectively applied in global rice mapping studies.

## 6 Data Availability

The 2023 East Asia 10 m annual rice distribution map can be accessed at Zenodo dataset from the following DOI: https://doi.org/10.5281/zenodo.13118409 (Song et al., 2024). The spatial reference system for this dataset is EPSG: 4326
(WGS84).

## 7 Conclusion

Addressing the sample generation challenges inherent in subcontinental-scale rice classification, in this paper, a novel large scale rice mapping framework was designed to generate the 10 m resolution rice distribution map of EA in 2023 (EARice10). The framework involves an indices-based sample set generation method and an optical-SAR adaptive fusion model, to take
full advantage of S1 and S2 data. The generated EARice10 has an OA of 90.48% on the validation samples, showing good consistency with statistical data and existing datasets, with $R^2$ values ranging between 0.94 and 0.98 with statistical data, and



between 0.79 and 0.98 with existing datasets. Moreover, EARice10 is the most up-to-date rice distribution map that comprehensively covers four rice production countries of EA in 10 m resolution. And for the first time, it covers rice areas of less than 100,000 ha in China, filling the data gaps of eight provincial administrative regions of China in previous studies.


**Author Contributions:** Conceptualization, methodology, software, M.S. and H.Z.; validation, formal analysis, J.G.; investigation, M.S. and H.Z.; resources, data curation, L.X. and F.W.; writing—original draft preparation, M.S., H.Z. and L.X.; writing—review and editing, H.Z., L.X., J.G. and L.Z.; visualization, J.J. ,Y.D., Y.X.; supervision, project administration, H.Z. and L.Z. All authors have read and agreed to the published version of the manuscript.

**Funding:** The research was supported by the International Research Centre of Big Data for Sustainable Development Goals (CBAS) [grant numbers CBAS2023SDG001], and the National Key R&D Program of China [grant numbers 2023YFB390620X).

**Acknowledgments:** The authors acknowledge the support of data and computational power provided by the Google Earth Engine platform.

**Conflicts of Interest:** The authors declare no conflict of interest.

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
