# Peer review of "EARice10: A 10 m Resolution Annual Rice Distribution Map of East Asia for 2023"

_Earth System Science Data, 2024_

## Author Response (AR1)

Dear Reviewers,

Manuscript ID essd-2024-331 entitled "A 10 m Resolution Annual Rice Distribution Map of East Asia for 2023."

We would like to express our sincere gratitude to the editor and both reviewers for their constructive feedback and thorough review of our manuscript. We have carefully considered all suggestions and have made the corresponding revisions to the manuscript. In addition to addressing the reviewers' comments, we have also refined the overall language to enhance the quality of the paper,and redrawn some of the figures for greater clarity. Below, we provide detailed responses to each of the reviewers' comments, including clarifications where necessary. We hope these revisions address the concerns and uncertainties raised by the reviewers. In the manuscript and this file, the blue parts are revisions suggested by the reviewer 1, green parts for suggestions of reviewer 2 are highlighted in green, and to improve the readability and overall quality of the paper, additional modifications are marked in red.

Sincerely,
Zhang Hong
zhanghong@radi.ac.cn

**Response to Reviewer 1**

**Comments to the Author**

**This paper maps rice paddy distribution of East Asia countries in 2023 at 10m resolution (EARice10), it is a timely and accurate (according to the assessment) product that can help estimate greenhouse gas emissions and grain yield. The paper developed a new method that integrates the use of both Sentinel-2 Optical and Sentinel-1 SAR data. The paper is in general well-written with high-quality figures. I have a few comments that I hope can help improve the manuscript.**

**RESPONSE:** Thank you very much for your appreciation of our work. Your suggestions have helped us a lot to improve the quality of our manuscript.

**1. My major concern is about the independent validation sample (91,320). The authors briefly mentioned they are obtained through field surveys and visual interpretations. How much is obtained through field surveys? How can visual interpretation accurately obtain validation samples? It needs details about visual interpretations and shows that it is robust. I doubt how one can visually distinguish rice paddies from other crops.**

**RESPONSE:** Thank you very much for your comment. Due to space limitations, we only briefly mentioned in the manuscript that the independent validation samples were obtained through field surveys and visual interpretation. In response to this concern, we provide a more detailed explanation here regarding the construction of the independent validation sample set.

[Figure]

[Figure]

[Figure]

[Figure]

**Figure R1-1. Field survey photos of rice paddies in different regions: (a) Heilongjiang; (b) Anhui; (c) Guangdong; (d) Tianjin.**

To evaluate the reliability of EARice10, we constructed an independent validation sample set consisting of 91,320 samples (covering a total of 4,384 rice and non-rice plots), spanning four East Asian countries (China, Japan, the Democratic People's Republic of Korea, and the Republic of Korea). Of these, 57,486 samples were collected in China, 15,834 in Japan, and 9,000 each in the Democratic People's Republic of Korea and the Republic of Korea. Due to varying accessibility across countries, different sampling strategies were employed. For China, we used a combination of field surveys and visual interpretation to collect validation samples. We conducted field surveys in China's major rice production areas, as shown in Figure R1-1. A total of 10,548 field samples (517 plots) were collected, including 5,346 rice samples (261 plots) and 5,202 non-rice samples (256 plots), accounting for 18.35% of the total samples in China and 11.56% of the total samples across East Asia. In Japan, the Democratic People's Republic of Korea, and the Republic of Korea, where field surveys were not feasible, we employed visual interpretation and cross-validation using multiple data sources to obtain validation samples.

Visual interpretation is a widely used method in remote sensing, particularly in cases where field surveys are not possible (Huang et al., 2023; Zhang et al., 2023; Li et al., 2024a; Li et al., 2024b). Before conducting visual interpretation, we utilized various reference materials for the study area, including news reports, statistical yearbooks, and land use and land cover (LULC) data. These resources provided valuable background information, enabling us to more accurately determine the overall distribution of rice in the study area. Based on this prior knowledge, we implemented strict guidelines for visual interpretation through cross-validation with multiple data

sources to ensure accuracy and robustness. The primary data sources used include:

**(1) Sentinel-2 imagery in 2023**

Numerous studies have shown that during specific rice growth periods, false-color composite images can effectively distinguish rice from other crops (Zhang et al., 2023; Li et al., 2024a; Li et al., 2024b). The "soil-water-vegetation" characteristics of rice paddies during the transplanting period exhibit distinct spectral features. Using SWIR (B11), NIR (B8), and Red (B4) band combinations to generate false-color composite images, rice paddies can be clearly differentiated from other crops (as shown in the left column of Figure R1-2). Generally, during the rice transplanting period, the surface of rice paddies is mostly submerged, resulting in low spectral values in the SWIR and NIR bands, appearing as dark green features in false-color composite images, while other crops typically appear red or brown.

**(2) Google high-resolution satellite imagery**

In Google high-resolution satellite imagery, rice paddies in East Asia typically appear as regular small grid-like plots due to the high irrigation requirements of rice, which are not characteristic of other dryland crops (Zhao et al., 2021; Li et al., 2024b). Although Google high-resolution satellite imagery is often historical, the annual variation in large rice-growing areas is relatively small. Therefore, it serves as a useful reference for distinguishing rice from other crops (as shown in the middle column of Figure R1-2).

**(3) Historical rice distribution maps**

As mentioned earlier, rice-growing areas in East Asia tend to remain relatively stable due to the high irrigation requirements of rice. Hence, historical rice distribution maps for the study area can serve as an effective means of identifying rice distribution (as shown in the right column of Figure R1-2).

By combining multiple data sources, we were able to significantly improve the accuracy and reliability of the independent validation sample set obtained through

visual interpretation. In areas with historical rice distribution maps, we used all three data sources (1), (2), and (3) for visual interpretation of rice and non-rice areas. In areas where historical maps were unavailable, we exclusively relied on data sources (1) and (2).

Figure R1-2 illustrates the reliability of visual interpretation using multiple data sources. The first row shows an overview of the entire study area, the second row presents zoomed-in "rice paddy areas," and the third row displays "non-rice crop areas." The combination of different data sources enables better differentiation between rice and other crops, ensuring the reliability of the independent validation sample set.

[Figure]

**Figure R1-2. Visual interpretation of rice and other crops using multiple data sources © Google Maps 2023.**

**2. Eq (5), it needs more details that how coefficient B and A are determined, based on what comparison and criteria. Are these empirically derived values B and A needs change when applied to a different region, for example South Asia?**

**If so, how others adopt this method can determine the values.**

**RESPONSE:** Thank you very much for your comment. We provide a detailed explanation here of how the empirical coefficients B and A in Equation (5) were determined.

Based on our previous research, the temporal statistical characteristics of the VH polarization from SAR data can effectively distinguish rice from other land cover types (Xu et al., 2021; Sun et al., 2023). Rice exhibits distinct ranges for various VH temporal statistical features (such as maximum, minimum, mean, and variance). To quantify the SRMI (SAR-based rice mapping index) into a rice probability, we normalize these four temporal features using Equation (5), where coefficients A and B are employed to define the lower and upper bounds for the normalized features. The criteria for determining A and B are to ensure that the SRMI values range between 0 and 1, while maintaining the ability to differentiate rice from other land cover types. The specific values of A and B were determined based on an extensive analysis of the VH temporal curves and statistical characteristics of rice from different regions, resulting in the most suitable empirical parameters for rice identification in East Asia.

Figures R1-3 to R1-8 illustrate the VH temporal curves and statistical features of rice across different countries and cropping systems in East Asia. As shown, the maximum backscatter coefficient ($\sigma_{max}^0$) for rice generally does not exceed -10 dB, and the closer the value is to -10 dB, the higher the probability that the plot is rice. The minimum backscatter coefficient ($\sigma_{min}^0$) for rice is typically around -25 dB. The mean backscatter coefficient ($\sigma_{mean}^0$) for rice usually ranges between -10 dB and -20 dB, with values closer to -20 dB indicating a higher probability of being rice. The variance of the backscatter coefficient ($\sigma_{var}^0$), for rice and other agricultural fields, is generally above -10. Based on this analysis of rice's temporal statistical features across different regions of East Asia, we determined that for $\sigma_{max}^0$ and $\sigma_{min}^0$, A is set to -25 and B to -10. For $\sigma_{mean}^0$, A is set to -20 and B to -10, and for $\sigma_{var}^0$, A is set to 0 and B to 10. After normalization using these parameters, Equation (6) yields SRMI values that mostly cluster around 0.6. With a threshold $\beta$ = 0.5, rice and other land

cover types can be effectively distinguished, as shown in Figure R1-9.

Furthermore, when applying these empirical parameters A and B to other regions, such as Southeast Asia, we found that the SRMI calculation also showed some effectiveness. Figures R1-10 and R1-11 present the results of rice mapping in Vietnam using this index, demonstrating that the threshold of 0.5 can still effectively distinguish rice from other land cover types. For regions outside of East Asia, we recommend fixing the empirical parameters A and B, while analyzing the specific distribution of SRMI for rice in that region to determine the most suitable threshold, thereby maximizing classification accuracy.

[Figure]

**Figure R1-3. VH temporal backscatter curves and statistical features of rice in Heilongjiang, China**

[Figure]

**Figure R1-4. VH temporal backscatter curves and statistical features of rice in the Democratic People's Republic of Korea**

[Figure]

**Figure R1-5. VH temporal backscatter curves and statistical features of rice in the Republic of Korea**

[Figure]

**Figure R1-6. VH temporal backscatter curves and statistical features of rice in Japan**

[Figure]

**Figure R1-7. VH temporal backscatter curves and statistical features of early rice in Guangdong, China**

[Figure]

**Figure R1-8. VH temporal backscatter curves and statistical features of late rice in Guangdong, China**

[Figure]

**Figure R1-9. SRMI values for different land cover types in East Asia**

[Figure]

**Figure R1-10. VH temporal backscatter curves and statistical features of rice in Vietnam**

[Figure]

| Google Satellite Imagery | Existing Rice Area Map | SRMI > 0.5 |

**Figure R1-11. Rice distribution mapping in Vietnam using SRMI**

**3. Line 203, Page 11: " divide the plots in to different objects", this sentence is very confusing. What are the plots referring to? Is there an image segmentation done or not? Do optical images and SAR images both go through the same procedure of image segmentation? It not sure if the final classification is based on pixels or a group of pixels- objects.**

**RESPONSE:** Thank you very much for your comment, and we apologize for the confusion caused by our inaccurate wording. The term "plots" was inappropriate here; we actually meant "dividing the image into different objects." We appreciate you pointing this out, and we have revised the original sentence to improve clarity. The specific modification is as follows:

**Line 205, Page 11**

...Subsequently, to mitigate the impact of SAR data speckle noise, the Simple Non-Iterative Clustering (SNIC) (Achanta and Susstrunk, 2017) superpixel segmentation was applied to divide the image into different objects...

In Section 3.1, Indices-based sample set generation method, only the Sentinel-1 SAR images underwent image segmentation, while the Sentinel-2 optical images were not segmented. The primary reason for segmenting SAR images was to reduce the speckle noise inherent in SAR data, and this segmentation process is described only in Section 3.1.1. In contrast, Sentinel-2 optical images do not have speckle noise issues,

so they were processed at the pixel level to retain spatial detail. In Section 3.2, where the classification model is discussed, the entire model operates at the pixel level, not at the object level.

**4. Line 205, Page 11, why size parameter of 15, compactness of 0.8, and connectivity of 8. It seems the SAR images are segmented; how do you treat the optical images then?**

**RESPONSE:** Thank you very much for your comment.

The parameters (size of 15, compactness of 0.8, and connectivity of 8) are the three hyperparameters of the SNIC (Simple Non-Iterative Clustering) superpixel segmentation algorithm. These parameters were empirically set to effectively balance noise and detail in the East Asia study area.

Given the characteristics of different data sources, we adopted different processing strategies during the sample generation step in Section 3.1. For SAR imagery (Sentinel-1), we applied superpixel segmentation to reduce the impact of noise. For optical imagery (Sentinel-2), we directly processed the data at the pixel level to better retain the details of the imagery.

Once again, we appreciate your valuable feedback, and we have revised Section 3.1.2 to enhance clarity:

**Line 230, Page 13**

...Unlike Sentinel-1 data, the operations for Sentinel-2 data were conducted at the pixel level....

**5. Line 236-237, page 13. What are these 2000 samples used for? Is it for training the following random forest model? Are they verified? I can see these 2000 samples have a very high probability of being rice if both Rice$_{Optical}$ and**

**Rice$_{SAR}$=1, but still no 100% guarantee.**

**RESPONSE:** Thank you very much for your comment. These 2000 samples are indeed used for training the random forest model, as shown in Figure 3 of the manuscript.

[Figure]

**Figure 3. The overall workflow of the proposed rice distribution mapping framework.**

As described in Section 3.1, we proposed an indices-based sample set generation

method, which creates a training sample set by combining Sentinel-2 optical data and Sentinel-1 SAR data. This method is intended to generate high-quality training samples when no historical rice distribution data or field survey data is available. These samples are used for the training of the optical-SAR adaptive fusion model described in Section 3.2.

For optical data, we used the LSWI and NDVI indices from Sentinel-2 to generate rice candidate areas. LSWI and NDVI have been widely used in large-scale rice mapping studies in Asia with high classification accuracy (Xiao et al., 2005; Xiao et al., 2006; Dong et al., 2016). However, relying solely on optical data in cloud-covered areas remains limited, leading to potential misclassifications or omissions of rice fields.

Considering that SAR can operate in all-weather conditions, existing research has demonstrated that using SAR data alone combined with phenological information can achieve high-accuracy rice mapping(Zhan et al., 2021; Xu et al., 2023). Therefore, we proposed the SRMI, an index that requires only one rice-growing season to map rice areas. Using this index, we generated SAR-based rice candidate areas, as demonstrated in Figure 5 of the manuscript, which shows the index's effectiveness in different countries and cropping backgrounds.

[Figure]

**Figure 5. Demonstrations of rice candidate areas in different regions based on SRMI.**

To minimize the uncertainty introduced by a single data source, we applied a strict sample selection criterion by combining the rice candidate areas generated from both optical and SAR data, as described in Equation (9). Rice samples were selected only in areas where both $\text{Rice}_{\text{Opitcal}} = 1$ and $\text{Rice}_{\text{SAR}} = 1$, while non-rice samples were collected in areas where both $\text{Rice}_{\text{Opitcal}} = 0$ and $\text{Rice}_{\text{SAR}} = 0$. This combined selection strategy effectively avoids misclassification or omission errors that could result from using a single data source, ensuring the high quality and reliability of the selected samples.

Although these 2000 samples were not independently validated, the results of the entire model were evaluated using an independent validation dataset in Section 4.2 (see Table 3 and Figure 14 of the manuscript). The validation results show that the

overall accuracy of EARice10 is 90.48%, with a user's accuracy of 90.93% and a producer's accuracy of 90.49%. These results validate the effectiveness of our sample generation strategy.

**Table 1.National-level confusion matrix for the EARice10 against validation sample set**

| Country | Class | Where **Rice*** and **Non-rice*** represent the validation sample points. | | | | |
| --- | --- | --- | --- | --- | --- | --- |
| | | **Rice*** | **Non-rice*** | **UA (%)** | **PA (%)** | **OA (%)** |
| China | Rice | 26736 | 2979 | 89.97% | 89.15% | 89.16% |
| | Non-rice | 3255 | 24516 | 88.28% | 89.17% | |
| Japan | Rice | 7152 | 816 | 89.76% | 90.34% | 90.02% |
| | Non-rice | 765 | 7101 | 90.27% | 89.69% | |
| The Republic of Korea | Rice | 4269 | 216 | 95.18% | 94.87% | 95.03% |
| | Non-rice | 231 | 4284 | 94.88% | 95.20% | |
| The Democratic People's Republic of Korea | Rice | 4290 | 222 | 95.08% | 95.33% | 95.20% |
| | Non-rice | 210 | 4278 | 95.32% | 95.07% | |
| East Asia | Rice | 42447 | 4233 | 90.93% | 90.49% | 90.48% |
| | Non-rice | 4461 | 40179 | 90.01% | 90.47% | |

[Figure]

**Figure 14. Provincial-level confusion matrix metrics for the EARice10 based on validation sample set**

**6. 3.2 Optical-SAR adaptive fusion model. This part, while it is creative, it is not necessary and seems too complicated to me. So about 24 RF models are trained for semi-month mean features of Sentinel-2 images. What are the input image features, are these the same as Table 1? What are the training labels (from 3.1.3?), are all RF models trained using the same set of training labels?**

**RESPONSE:** Thank you very much for your comment.

We appreciate your recognition of the model's creativity. Although the model involves multiple RF classifiers, its actual operational complexity is relatively low due to the parallel computing capabilities of the GEE cloud platform.

In this study, we only used Sentinel-1 and Sentinel-2 data from the rice-growing season for training. Specifically, in the single-season rice areas of northern regions,

the growing season lasts from early May to late September, with around 10 RF models trained on semi-monthly composite Sentinel-2 images. In the double-season rice areas of southern regions, the growing season runs from early March to late October, resulting in 16 RF models trained on semi-monthly composite images.

The input features consist of Sentinel-2 and Sentinel-1 image features. For Sentinel-2 data, we selected 10 original bands (including blue, green, red, near-infrared (NIR), red-edge (RE) bands 1-4, and short-wave infrared (SWIR) bands 1-2), along with several commonly used spectral indices (as listed in Table 1). For Sentinel-1 data, the input features included VH polarization, VV polarization, and the cross-polarization ratio (CR).

The training labels were obtained from the training samples generated in Section 3.1.3, and all RF models within the optical-SAR adaptive fusion model were trained using the same set of training labels. During the training process, cloud-covered areas in the Sentinel-2 images were marked as NoData after cloud removal. GEE automatically ignores these NoData pixels. Therefore, although all RF models use the same training labels, the number of effective samples varies due to cloud cover differences in different semi-monthly images.

**6. When a model is trained on each semi-month mean image S-2, you lose the temporal dynamic information as shown in Figure 6. I am not sure of the reasoning behind doing this, or whether it has any advantages. Second, why would the model bother running on cloudy pixels, why not just masking cloudy pixels out?**

**RESPONSE:** Thank you very much for your comment. We provide a detailed explanation of the reasoning behind the model design in Section 3.2.

Traditional methods typically classify time-series features extracted from multi-temporal Sentinel-2 images using a single RF model, which requires each pixel to have complete observations across the entire time series. In these conventional rice

mapping methods, cloud-affected regions are interpolated after cloud removal to ensure each pixel has a full time-series (You and Dong, 2020; You et al., 2021; Zhang et al., 2023). However, in areas with severe cloud cover (such as coastal regions or southern China), cloud distribution is highly variable in both time and space, and interpolation methods often fail to accurately restore spectral time-series features, potentially introducing unrealistic information that negatively impacts classification accuracy.

Our proposed optical-SAR adaptive fusion model is designed to leverage the respective strengths of optical and SAR data. While temporal dynamic information from Sentinel-2 might be incomplete in certain time periods, the spectral features of single time points can still provide sufficient information to distinguish rice from other land cover types (Zhang et al., 2021; Zhang et al., 2023). Therefore, after removing the cloud effects, we did not interpolate cloud-covered areas. Instead, multiple RF models were trained separately on semi-monthly composite Sentinel-2 images. The results of the different semi-monthly RF models were then combined through weighted voting (Equation 10), allowing us to build a rice classification probability map without relying on complete time-series information.

The advantages of the optical-SAR adaptive fusion model designed for Sentinel-2 data are as follows:

(1) **Avoiding interpolation errors:** Interpolation in highly cloud-covered areas is often inaccurate. Our model directly ignores NoData values in these areas, avoiding the introduction of erroneous information.

(2) **Maximizing the use of optical features**: Even without complete temporal dynamic information, the rich spectral features of Sentinel-2 provide enough information for classification at single time points. By voting across different time points, we improve classification accuracy.

(3) **Avoiding the "Hughes phenomenon":** Using multi-temporal data can lead to the "Hughes phenomenon," where classification accuracy decreases due to overly high

feature dimensionality. Our model avoids this by processing each time point separately.

**8. How is the model trained to combine $P_{Optical}$, $N_{cloud-free}$, and $P_{SAR}$? Again, training labels.**

**RESPONSE:** Thank you very much for your comment. Thank you very much for your comment. We will explain the training process of combining $P_{Optical}$, $N_{Cloud-free}$, and $P_{SAR}$.

[Figure]

**Figure 7. Structure of the optical-SAR adaptive fusion model using optical and SAR data.**

As shown in Figure 7, we used a separate random forest (RF) model to combine $P_{Optical}$, $N_{Cloud-free}$, and $P_{SAR}$, aiming to further explore the potential relationships

between the rice identification results from the two data sources. During the training process, this model's training labels are the same as those generated in Section 3.1.3. Additionally, all models in the 3.2 Optical-SAR adaptive fusion model are trained using this same set of labels.

Throughout Section 3.2, the Optical-SAR adaptive fusion model relies on the training samples generated in Section 3.1.3. By integrating the rich spectral information from optical data with the all-weather monitoring capability of SAR data, we were able to construct a more robust classification model for high-precision rice mapping.

**9. Line 346, Page 20. This refers to my previous comments, more details are needed to convince the independent samples are reliable.**

**RESPONSE:** Thank you very much for your comment. As we mentioned in the response to Question 1, the independent validation samples were generated through a combination of field surveys and rigorous visual interpretation. In cases where a large amount of field survey data was unavailable, visual interpretation is a widely used and reliable method for expanding datasets, as demonstrated in many related studies (Zhang et al., 2023; Li et al., 2024a; Li et al., 2024b). During the visual interpretation process, we combined multiple data sources, including Sentinel-2 optical imagery, Google high-resolution imagery, and existing historical rice distribution maps. Cross-validation among these data sources ensured the accuracy of the interpretation results. Through this multi-source validation and cross-comparison, we were able to construct a high-quality independent validation sample set, ensuring its reliability in both spatial distribution and land cover type.

**Response to Reviewer 2**

**Comments to the Author**

**This paper maps rice paddy distribution of East Asia countries in 2023 at 10m resolution (EARice10), a timely and accurate product (according to the assessment) that is useful for estimating greenhouse gas emissions and grain yield. The method integrates both Sentinel-2 Optical and Sentinel-1 SAR data. The manuscript is generally well-written, with high-quality figures. However, I have a few concerns that I hope will help improve the manuscript.**

**RESPONSE:** Thank you very much for your appreciation of our work. Your suggestions have helped us a lot to improve the quality of our manuscript.

**1. Validation of Data**

**The independent validation samples (91,320) are said to be obtained through field surveys and visual interpretations. How many of these samples were obtained through field surveys?**

**RESPONSE:** Thank you very much for your comment. Due to space limitations, we only briefly mentioned in the manuscript that the independent validation samples were obtained through field surveys and visual interpretation. In response to this concern, we provide a more detailed explanation here regarding the construction of the independent validation sample set.

[Figure]

[Figure]

[Figure]

[Figure]

**Figure R2-1. Field survey photos of rice paddies in different regions: (a) Heilongjiang; (b) Anhui; (c) Guangdong; (d) Tianjin.**

To evaluate the reliability of EARice10, we constructed an independent validation sample set consisting of 91,320 samples (covering a total of 4,384 rice and non-rice plots), spanning four East Asian countries (China, Japan, the Democratic People's Republic of Korea, and the Republic of Korea). Of these, 57,486 samples were collected in China, 15,834 in Japan, and 9,000 each in the Democratic People's Republic of Korea and the Republic of Korea. Due to varying accessibility across countries, different sampling strategies were employed. For China, we used a combination of field surveys and visual interpretation to collect validation samples. We conducted field surveys in China's major rice production areas, as shown in Figure R2-1. Through these field surveys, a total of 10,548 field samples (517 plots) were collected, including 5,346 rice samples (261 plots) and 5,202 non-rice samples (256 plots), accounting for 18.35% of the total samples in China and 11.56% of the total samples across East Asia. In Japan, the Democratic People's Republic of Korea, and the Republic of Korea, where field surveys were not feasible, we employed visual interpretation and cross-validation using multiple data sources to obtain validation samples.

Visual interpretation is a widely used method in remote sensing, particularly in cases where field surveys are not possible (Huang et al., 2023; Zhang et al., 2023; Li et al., 2024a; Li et al., 2024b). Before conducting visual interpretation, we utilized various reference materials for the study area, including news reports, statistical yearbooks, and land use and land cover data. These resources provided valuable background information, enabling us to more accurately determine the overall distribution of rice in the study area. Based on this prior knowledge, we implemented strict guidelines for

visual interpretation through cross-validation with multiple data sources to ensure accuracy and robustness. The primary data sources used include:

**(1) Sentinel-2 imagery from 2023:**

Numerous studies have shown that during specific rice growth periods, false-color composite images can effectively distinguish rice from other crops (Zhang et al., 2023; Li et al., 2024a; Li et al., 2024b). The "soil-water-vegetation" characteristics of rice paddies during the transplanting period exhibit distinct spectral features. Using SWIR (B11), NIR (B8), and Red (B4) band combinations to generate false-color composite images, rice paddies can be clearly differentiated from other crops (as shown in the left column of Figure R2-2). Generally, during the rice transplanting period, the surface of rice paddies is mostly submerged, resulting in low spectral values in the SWIR and NIR bands, appearing as dark green features in false-color composite images, while other crops typically appear red or brown.

**(2) Google high-resolution satellite imagery:**

In Google high-resolution satellite imagery, rice paddies in East Asia typically appear as regular small grid-like plots due to the high irrigation requirements of rice, which are not characteristic of other dryland crops (Zhao et al., 2021; Li et al., 2024b). Although Google high-resolution satellite imagery is often historical, the annual variation in large rice-growing areas is relatively small. Therefore, it serves as a useful reference for distinguishing rice from other crops (as shown in the middle column of Figure R2-2).

**(3) Historical rice distribution maps:**

As mentioned earlier, rice-growing areas tend to remain relatively stable due to the high irrigation requirements of rice in East Asia. Hence, historical rice distribution maps for the study area can serve as an effective means of identifying rice distribution (as shown in the right column of Figure R2-2).

By combining multiple data sources, we were able to significantly improve the

accuracy and reliability of the independent validation sample set obtained through visual interpretation. In areas with historical rice distribution maps, we used all three data sources (1), (2), and (3) for visual interpretation of rice and non-rice areas. In areas where historical maps were unavailable, we exclusively relied on data sources (1) and (2).

Figure R2-2 illustrates the reliability of visual interpretation using multiple data sources. The first row shows an overview of the entire study area, the second row presents zoomed-in "rice paddy areas," and the third row displays "non-rice crop areas." The combination of different data sources enables better differentiation between rice and other crops, ensuring the reliability of the independent validation sample set.

[Figure]

**Figure R2-2. Visual interpretation of rice and other crops using multiple data sources © Google Maps 2023.**

**2. Application to Fragmented Rice Fields (Southern China)**

**Given that the method uses an optical-SAR adaptive fusion model, it may**

**struggle to perform well in fragmented rice fields, especially in mountainous southern China. The small, scattered fields and complex terrain in this region could lead to errors in classification.**

**It is recommended to incorporate additional geographical data (e.g., terrain, soil types) to limit the model and improve its performance in such areas. The model's adaptability to these factors could significantly enhance its accuracy in more complex regions.**

**RESPONSE:** Thank you very much for your comment. We appreciate your valuable suggestion. It is a common observation that existing models and open datasets perform better in plain areas compared to fragmented rice fields in mountainous southern China, which is a well-documented phenomenon in existing research (Dong and Xiao, 2016; Pan et al., 2021; Huang et al., 2023; Shen et al., 2023).

In our study, we also incorporated DEM data to analyze the experimental results, as shown in Section 4.1 of the manuscript, which presents the distribution of rice across different slopes and elevations in East Asia.

As you mentioned, incorporating high-resolution soil type data could indeed improve the accuracy of rice mapping. However, current soil type data are generally available at coarser scales, lacking sufficient spatial detail to meet the requirements of 10-meter resolution rice mapping (Liu et al., 2022; Nachtergaele et al., 2023).

Once again, we sincerely appreciate your suggestion. Should higher-resolution or more accurate soil type data become available in the future, we will actively explore incorporating such data into our research to further enhance the model's performance.

**3. Comparison with Existing Data**

**While the manuscript emphasizes the improved resolution of the rice map (10m), there are no comparisons made with prior datasets, particularly those from 2015-2021, which were at 500m resolution.**

**The authors should consider comparing the distribution and precision of the current data with these earlier datasets to demonstrate the advancements achieved through EARice10. This would help showcase the superior results of this study in a more concrete manner.**

**RESPONSE:** Thank you very much for your comment. We may not have been sufficiently detailed in our previous description. As shown in Figure 17 of the manuscript, we compared the spatial distribution of our dataset with an existing 500-meter dataset (APRA500 (500m) (Han et al., 2022). The results indicate that the 10-meter resolution dataset offers a significant improvement in spatial detail.

Due to the severe mixed-pixel phenomenon in the 500-meter resolution imagery, direct quantitative comparison may not be entirely fair. Therefore, we primarily conducted a visual comparison to illustrate the differences. To more clearly demonstrate these differences, we have provided additional spatial distribution comparison maps (see Figure R2-3). From these maps, it is evident that EARice10 (10m resolution) shows substantial improvements in spatial detail and accuracy compared to APRA500 (500m), particularly in terms of field edges and the distribution of small-scale rice paddies.

[Figure]

**Figure 17. Detailed comparison of the EARice10 with five existing rice distribution maps. Site 1 and Site 2 are single-season rice planting areas in China (Heilongjiang and Liaoning Province), Site 3 is a mixed-season rice planting area in China (Hubei Province), Site 4 is a double-season rice planting area in China (Guangdong Province), and Sites 5-7 are single-season rice planting areas located in the Democratic People's Republic of Korea, the Republic of Korea, and Japan, respectively.**

[Figure]

**Figure R2-3. Detail comparison of EARice10 (10m) and APRA500 (500m) datasets.**

**4. Specific Suggestions**

**In Equation (5), where the SAR features are normalized, the manuscript does**

**not clearly explain how the preset parameters A and B were derived. Are these values empirical, and if so, how were they selected? The paper should provide more justification or sensitivity analysis showing how different parameter values affect the results, particularly in regions like southern China where terrain and cropping patterns differ from other areas.**

**Additionally, if these parameters were tuned based on a specific region, would they need to be regionally adjusted when applying the method to other areas (e.g., South or Southeast Asia)? Clarifying this would improve the reproducibility of the approach.**

**RESPONSE:** Thank you very much for your comment. Thank you very much for your comment. We provide a detailed explanation here of how the empirical coefficients B and A in Equation (5) were determined.

Based on our previous research, the temporal statistical characteristics of the VH polarization from SAR data can effectively distinguish rice from other land cover types (Xu et al., 2021; Sun et al., 2023). Rice exhibits distinct ranges for various VH temporal statistical features (such as maximum, minimum, mean, and variance). To quantify the SRMI (SAR-based rice mapping index) into a rice probability, we normalize these four temporal features using Equation (5), where coefficients A and B are employed to define the lower and upper bounds for the normalized features. The criteria for determining A and B are to ensure that the SRMI values range between 0 and 1, while maintaining the ability to differentiate rice from other land cover types. The specific values of A and B were determined based on an extensive analysis of the VH temporal curves and statistical characteristics of rice from different regions, resulting in the most suitable empirical parameters for rice identification in East Asia.

Figures R2-4 to R2-9 illustrate the VH temporal curves and statistical features of rice across different countries and cropping systems in East Asia. As shown, the maximum backscatter coefficient ($\sigma_{max}^0$) for rice generally does not exceed -10 dB, and the closer the value is to -10 dB, the higher the probability that the plot is rice. The

minimum backscatter coefficient ($\sigma^0_{min}$) for rice is typically around -25 dB. The mean backscatter coefficient ($\sigma^0_{mean}$) for rice usually ranges between -10 dB and -20 dB, with values closer to -20 dB indicating a higher probability of being rice. The variance of the backscatter coefficient ($\sigma^0_{var}$), for rice and other agricultural fields, is generally above -10. Based on this analysis of rice's temporal statistical features across different regions of East Asia, we determined that for $\sigma^0_{max}$ and $\sigma^0_{min}$, A is set to -25 and B to -10. For $\sigma^0_{mean}$, A is set to -20 and B to -10, and for $\sigma^0_{var}$, A is set to 0 and B to 10. After normalization using these parameters, Equation (6) yields SRMI values that mostly cluster around 0.6. With a threshold $\beta$ = 0.5, rice and other land cover types can be effectively distinguished, as shown in Figure R2-10.

Furthermore, when applying these empirical parameters A and B to other regions, such as Southeast Asia, we found that the SRMI calculation also showed some effectiveness. Figures R2-11 and R2-12 present the results of rice mapping in Vietnam using this index, demonstrating that the threshold of 0.5 can still effectively distinguish rice from other land cover types. For regions outside of East Asia, we recommend fixing the empirical parameters A and B, while analyzing the specific distribution of SRMI for rice in that region to determine the most suitable threshold, thereby maximizing classification accuracy.

[Figure]

**Figure R2-4. VH temporal backscatter curves and statistical features of rice in Heilongjiang, China**

[Figure]

**Figure R2-5. VH temporal backscatter curves and statistical features of rice in the Democratic People's Republic of Korea**

[Figure]

**Figure R2-6. VH temporal backscatter curves and statistical features of rice in the Republic of Korea**

[Figure]

**Figure R2-7. VH temporal backscatter curves and statistical features of rice in Japan**

[Figure]

**Figure R2-8. VH temporal backscatter curves and statistical features of early rice in Guangdong, China**

[Figure]

**Figure R2-9. VH temporal backscatter curves and statistical features of late rice in Guangdong, China**

[Figure]

**Figure R2-10. SRMI values for different land cover types in East Asia**

[Figure]

**Figure R2-11. VH temporal backscatter curves and statistical features of rice in Vietnam**

[Figure]

| Google Satellite Imagery | Existing Rice Area Map | SRMI > 0.5 |

**Figure R2-12. Rice distribution mapping in Vietnam using SRMI**

**5. SRMI Threshold**

**The SRMI threshold for rice classification (set at 0.5) may not be flexible enough to handle the variability in terrain and crop conditions, particularly in fragmented and mountainous regions. It would be useful to conduct a sensitivity analysis on this threshold to see how changes in SRMI values affect classification accuracy in different landscapes. This analysis could highlight the robustness of the SRMI index across diverse geographic regions.**

**RESPONSE:** Thank you very much for your comment.

As shown in the box plot in Figure 4 of the manuscript, the SRMI values are distributed differently among various land cover types, and the results indicate that a threshold of 0.5 can effectively distinguish rice from other land cover types. Additionally, we performed extensive testing and validation of rice fields in multiple countries (e.g., Heilongjiang and Guangdong in China, Japan, the Republic of Korea, and the Democratic People's Republic of Korea) and across different landscapes, including plains and mountainous regions. The results in Figure 5 of the manuscript clearly demonstrate the application of SRMI in different regions. Although classification accuracy varies depending on terrain conditions, the threshold of 0.5 remains consistently effective in distinguishing rice from non-rice plots in most cases, showcasing the robustness of the SRMI index.

[Figure]

**Figure 4. The box plot of the SRMI value of land cover types.**

[Figure]

**Figure 5. Demonstrations of rice candidate areas in different regions based on SRMI.**

**6. Random Forest Model Training**

In Section 3.2, where multiple Random Forest models are trained for each semi-monthly image, there is a risk of over-complicating the model and losing temporal dynamics, as indicated by Figure 6. The paper should clarify why it was necessary to train multiple models rather than aggregating data over a longer period to maintain temporal information.

This section could also benefit from a discussion on overfitting risks, particularly in fragmented regions where training data may be more heterogeneous.

It is recommended to test a more streamlined approach that reduces the number

**of models and retains more temporal information to potentially improve classification performance, especially in areas with sparse data coverage.**

**RESPONSE:** Thank you very much for your comment.

Traditional methods typically classify time-series features extracted from multi-temporal Sentinel-2 images using a single RF model, which requires each pixel to have complete observations across the entire time series. In past rice mapping methods, cloud-affected areas were interpolated after cloud removal to ensure that each pixel had a complete time series (You and Dong, 2020; You et al., 2021; Zhang et al., 2023). However, in regions with severe cloud cover (e.g., coastal areas or southern China), the spatial and temporal distribution of clouds is highly variable (as shown in the red squares in Figure 7). Interpolation methods often fail to fully restore spectral time-series features, and may introduce inaccurate information that negatively impacts classification accuracy.

[Figure]

**Figure 7. Structure of the optical-SAR adaptive fusion model using optical and**

**SAR data.**

Our proposed optical-SAR adaptive fusion model is designed to leverage the respective strengths of optical and SAR data. Even though temporal dynamic information from Sentinel-2 may be incomplete at certain time points, its spectral features at individual time points can still provide enough information to distinguish rice from other land cover types (Zhang et al., 2021; Zhang et al., 2023). Therefore, after removing cloud effects, we did not interpolate cloud-covered areas. Instead, we trained multiple RF models for each semi-monthly composite Sentinel-2 image. The results of the RF models were then combined through weighted voting (Equation 10), allowing us to generate rice classification probabilities based on optical data without relying on a complete time series.

The advantages of the optical-SAR adaptive fusion model designed for Sentinel-2 data are as follows:

(1) **Avoiding interpolation errors**: Interpolation in cloud-covered areas is often inaccurate. Our model directly ignores NoData values in these regions, avoiding the introduction of artificial temporal information.

(2) **Maximizing the use of optical features**: Despite incomplete temporal dynamic information, the rich spectral features of Sentinel-2 provide sufficient information for accurate classification, even at individual time points. By voting on outputs from different time points, we further improve classification accuracy.

(3) **Avoiding the impact of the "Hughes phenomenon"**: Multi-temporal data with high feature dimensionality can lead to the "Hughes phenomenon," where classification accuracy decreases as dimensionality increases. Our model avoids this problem by processing each time point separately.

To validate the advantages of the optical-SAR adaptive fusion model in real-world applications, we selected Shanwei City in Guangdong Province, southern China, as a study area. This region is heavily affected by cloud cover and presents a challenging classification environment, making it an ideal test case for evaluating the model's

robustness in difficult conditions.

We compared four different approaches:

(1) **Our Model (S1+S2)**: An integrated model combining optical (Sentinel-2) and SAR (Sentinel-1) data without interpolating cloud-covered areas.

(2) **RF (S1)**: A random forest model trained using only Sentinel-1 SAR time-series data.

(3) **RF (S2)**: A random forest model trained using only Sentinel-2 optical time-series data, with cloud-covered areas interpolated.

(4) **RF (S1+S2)**: A random forest model combining Sentinel-1 and interpolated Sentinel-2 time-series data.

As shown in Figure R2-13, our model outperformed the other three RF models in terms of user's accuracy (91.89%), producer's accuracy (90.67%), and overall accuracy (91.33%). This demonstrates that combining optical and SAR data can significantly improve classification performance, especially in regions heavily affected by cloud cover.

[Figure]

**Figure R2-13. Comparison of the Optical-SAR adaptive fusion model with individual RF models.**

**7. Validation and Field Survey Data**

**The validation data is largely based on visual interpretation. However, the manuscript lacks details on how this interpretation was conducted, especially in regions with fragmented rice fields. Providing more concrete evidence, such as field survey data or ground truth photographs, would strengthen the validation process.**

**The authors should also clarify the proportion of samples that were obtained through actual field surveys versus visual interpretation.**

**Furthermore, it would be useful to provide accuracy metrics for different regions, especially comparing flat and mountainous areas, to demonstrate the model's** robustness across diverse landscapes.

**RESPONSE:** Thank you very much for your comment.

As mentioned in Response 1, our independent validation sample set was generated through a combination of field surveys and rigorous visual interpretation. The visual interpretation was cross-validated using multiple data sources (e.g., Sentinel-2 optical imagery, Google high-resolution imagery, and existing historical rice distribution maps). Visual interpretation is a widely applied and reliable method when field survey data is limited, and many related studies have validated its effectiveness and accuracy in rice distribution mapping (Zhang et al., 2023; Li et al., 2024a; Li et al., 2024b). Through multi-source data validation and cross-comparison, we were able to obtain high-quality independent validation samples, ensuring their accuracy in both spatial distribution and land cover type. Thus, these independent validation samples effectively verified the reliability of the final mapping results.

Field survey points accounted for approximately 18.35% of the total samples in China and 11.56% of the total samples across East Asia. Additionally, ground truth photos of rice fields from Heilongjiang, Anhui, and Guangdong were provided to further demonstrate the reliability of the sample set.

Once again, thank you for pointing out this issue. We have revised Section 2.3 to further improve clarity:

...To assess the accuracy of the generated rice map, an independent validation sample set containing 91,320 samples (46,908 rice and 44,412 non-rice) was constructed through field surveys and visual interpretation, with field survey points accounting for 11.56% of the total samples, as shown in Figure 2...

[Figure]

Figure 2.The Distribution of the validation sample set: (a, c, e) Sentinel-2 false-color images (R: SWIR1, G: NIR, B: Red); (b, d, f) enlarged local views of (a, c, e) respectively; (g) the distribution of rice and non-rice validation sample set. (h, i, j) ground truth photos of rice fields. Basemap sources for (g): Esri, TomTom, Garmin, FAO, NOAA, USGS, © OpenStreetMap contributors, and the GIS User Community.

Due to the lack of detailed terrain distribution data for East Asia, we followed the classification of cultivated land slopes specified by the Technical Regulations for the Third National Land Survey of China published by the Ministry of Natural Resources of China. Regions with slopes≤2° were defined as plains, and corresponding rice fields were classified as flatland rice cultivation; regions with slopes>2°were defined as mountainous rice fields (including terraced and sloped areas). Figures R2-14 and

R2-15 show the proportion of rice area and classification accuracy under different slopes across various countries.

As seen in Figure R2-15, while rice classification accuracy in flat areas is higher than in mountainous regions, the accuracy for all countries across different terrains remains above 88%, demonstrating the model's robustness in diverse geographic conditions.

[Figure]

**Figure R2-14. Proportion of rice area under different slopes in various countries.**

[Figure]

**Figure R2-15. Classification accuracy under different slopes in various countries.**

**References**

Achanta, R. and Susstrunk, S.: Superpixels and polygons using simple non-iterative clustering, Proceedings of the IEEE conference on computer vision and pattern recognition, 4651-4660,

Dong, J. and Xiao, X.: Evolution of regional to global paddy rice mapping methods: A review, ISPRS Journal of Photogrammetry and Remote Sensing, 119, 214-227, 10.1016/j.isprsjprs.2016.05.010, 2016.

Dong, J., Xiao, X., Menarguez, M. A., Zhang, G., Qin, Y., Thau, D., Biradar, C., and Moore, B., 3rd: Mapping paddy rice planting area in northeastern Asia with Landsat 8 images, phenology-based algorithm and Google Earth Engine, Remote Sens Environ, 185, 142-154, 10.1016/j.rse.2016.02.016, 2016.

Han, J., Zhang, Z., Luo, Y., Cao, J., Zhang, L., Zhuang, H., Cheng, F., Zhang, J., and Tao, F.: Annual paddy rice planting area and cropping intensity datasets and their dynamics in the Asian monsoon region from 2000 to 2020, Agricultural Systems, 200, 10.1016/j.agsy.2022.103437, 2022.

Huang, C., You, S., Liu, A., Li, P., Zhang, J., and Deng, J.: High-Resolution National-Scale Mapping of Paddy Rice Based on Sentinel-1/2 Data, Remote Sensing, 15, 10.3390/rs15164055, 2023.

Li, H., Huang, J., Zhang, C., Ning, X., Zhang, S., and Atkinson, P. M.: An efficient and generalisable approach for mapping paddy rice fields based on their unique spectra during the transplanting period leveraging the CIE colour space, Remote Sensing of Environment, 313, 114381, 2024a.

Li, L., Zhou, D., Liu, K., Shi, T., Xie, C., Wang, S., Li, H., Dong, G., and Li, X.: Optimizing Rice Field Mapping in the Northern Region of China: An Asynchronous Flooding Signal and Object-Based Method, IEEE Journal of Selected Topics in Applied Earth Observations and Remote Sensing, 1-16, 10.1109/jstars.2024.3357141, 2024b.

Liu, F., Wu, H., Zhao, Y., Li, D., Yang, J.-L., Song, X., Shi, Z., Zhu, A.-X., and Zhang, G.-L.: Mapping high resolution national soil information grids of China, Science Bulletin, 67, 328-340, 2022.

Nachtergaele, F., van Velthuizen, H., Verelst, L., Wiberg, D., Henry, M., Chiozza, F., Yigini, Y., Aksoy, E., Batjes, N., and Boateng, E.: Harmonized world soil database version 2.0, FAO2023.

Pan, B., Zheng, Y., Shen, R., Ye, T., Zhao, W., Dong, J., Ma, H., and Yuan, W.: High Resolution Distribution Dataset of Double-Season Paddy Rice in China, Remote Sensing, 13, 10.3390/rs13224609, 2021.

Shen, R., Pan, B., Peng, Q., Dong, J., Chen, X., Zhang, X., Ye, T., Huang, J., and Yuan, W.: High-resolution distribution maps of single-season rice in China from 2017 to 2022, Earth System Science Data, 15, 3203-3222, 10.5194/essd-15-3203-2023, 2023.

Sun, C., Zhang, H., Xu, L., Ge, J., Jiang, J., Zuo, L., and Wang, C.: Twenty-meter annual paddy rice area map for mainland Southeast Asia using Sentinel-1 synthetic-aperture-radar data, Earth System Science Data, 15, 1501-1520, 10.5194/essd-15-1501-2023, 2023.

Xiao, X., Boles, S., Frolking, S., Li, C., Babu, J. Y., Salas, W., and Moore III, B.: Mapping paddy rice agriculture in South and Southeast Asia using multi-temporal MODIS images, Remote sensing of Environment, 100, 95-113, 10.1016/j.rse.2005.10.004, 2006.

Xiao, X., Boles, S., Liu, J., Zhuang, D., Frolking, S., Li, C., Salas, W., and Moore III, B.: Mapping paddy rice agriculture in southern China using multi-temporal MODIS images, Remote sensing of environment, 95, 480-492, https://doi.org/10.1016/j.rse.2004.12.009, 2005.

Xu, L., Zhang, H., Wang, C., Wei, S., Zhang, B., Wu, F., and Tang, Y.: Paddy Rice Mapping in Thailand Using Time-Series Sentinel-1 Data and Deep Learning Model, Remote Sensing, 13, 3994, 10.3390/rs13193994, 2021.

Xu, S., Zhu, X., Chen, J., Zhu, X., Duan, M., Qiu, B., Wan, L., Tan, X., Xu, Y. N., and Cao, R.: A robust index to extract paddy fields in cloudy regions from SAR time series, Remote Sensing of Environment, 285, 10.1016/j.rse.2022.113374, 2023.

You, N. and Dong, J.: Examining earliest identifiable timing of crops using all available Sentinel 1/2 imagery and Google Earth Engine, ISPRS Journal of Photogrammetry and Remote Sensing, 161, 109-123, 10.1016/j.isprsjprs.2020.01.001, 2020.

You, N., Dong, J., Huang, J., Du, G., Zhang, G., He, Y., Yang, T., Di, Y., and Xiao, X.: The 10-m crop type maps in Northeast China during 2017–2019, Scientific data, 8, 41, 2021.

Zhan, P., Zhu, W., and Li, N.: An automated rice mapping method based on flooding signals in synthetic aperture radar time series, Remote Sensing of Environment, 252, 10.1016/j.rse.2020.112112, 2021.

Zhang, C., Zhang, H., and Tian, S.: Phenology-assisted supervised paddy rice mapping with the Landsat imagery on Google Earth Engine: Experiments in Heilongjiang Province of China from 1990 to 2020, Computers and Electronics in Agriculture, 212, 108105, 2023.

Zhang, C., Zhang, H., and Zhang, L.: Spatial domain bridge transfer: An automated paddy rice mapping method with no training data required and decreased image inputs for the large cloudy area, Computers and Electronics in Agriculture, 181, 105978, 2021.

Zhao, R., Li, Y., and Ma, M.: Mapping Paddy Rice with Satellite Remote Sensing: A Review, Sustainability, 13, 10.3390/su13020503, 2021.

---

## Author Response (AR2)

Dear Editor and Reviewers,

Manuscript ID essd-2024-331 entitled "A 10 m Resolution Annual Rice Distribution Map of East Asia for 2023."

We would like to express our sincere gratitude to the editor and both reviewers for their constructive feedback and thorough review of our manuscript. We have carefully considered all suggestions and have made the corresponding revisions to the manuscript. In addition to addressing the reviewers' comments, we have also refined the overall language to enhance the quality of the paper, and redrawn some of the figures for greater clarity. Below, we provide detailed responses to each of the editor's and reviewers' comments, including clarifications where necessary. We hope these revisions address the concerns and uncertainties raised by the reviewers. In the manuscript and this file, the blue parts are revisions suggested by the reviewer #3, green parts for suggestions of reviewer #4 are highlighted in green, and to improve the readability and overall quality of the paper, additional modifications are marked in red.

Sincerely,

Zhang Hong

zhanghong@radi.ac.cn

**Response to the Editor**

**Comments to the Author**

**Please address the reviewers' comments thoroughly and ensure that the formatting of both the data and the manuscript aligns with ESSD's requirements.**

**RESPONSE:** Thank you very much for your valuable suggestions. We have thoroughly addressed all comments from the reviewers by providing detailed responses to each point and making corresponding revisions to the manuscript. Additionally, we have carefully reviewed the overall structure and formatting of the manuscript to ensure full compliance with ESSD's requirements.

We sincerely appreciate your guidance and have submitted the revised manuscript accordingly.

**Response to Reviewer #2**

**RESPONSE:** Thank you very much for recognizing our work and for your valuable feedback on our manuscript during the review process. Your constructive comments and insightful suggestions have significantly contributed to improving the quality of our study.

We sincerely appreciate your support and are grateful for the opportunity to have our manuscript reviewed by you. Your acceptance of our work is a tremendous encouragement to our team. Thank you once again for your time and effort.

**Response to Reviewer #3**

**Comments to the Author**

**This study presents a valuable approach for high-resolution annual rice distribution mapping in East Asia using Google Earth Engine (GEE). The novel Synthetic Aperture Radar (SAR)-based Rice distribution Mapping Index (SRMI) and stacking-based optical-SAR adaptive fusion model demonstrate high accuracy and reliability, achieving an overall accuracy of 90.48%. The open access data and availability of the product on Zenodo promote further research and utilization. While the study's focus is on East Asia, exploring its adaptability to different regions and integrating additional data sources could further enhance its potential. Analyzing trends over multiple years would provide valuable insights into changes in rice production and its impact on food security and the environment.**

**The authors have also incorporated the previous reviewers' comments in the revision. I have some minor comments before its acceptance for publication.**

**RESPONSE:** Thank you very much for your recognition and acknowledgment of our work. We greatly appreciate your positive evaluation and constructive suggestions, which have greatly contributed to improving the quality of our manuscript. We have carefully addressed all your comments and made corresponding revisions to further enhance the clarity and completeness of the paper.

**1. Line 158-167: these detailed statements on the existing products can be removed to avoid repetition.**

**RESPONSE:** Thank you very much for your suggestion. We have removed the detailed statements in Lines 158-167 to avoid repetition and improve the conciseness of the manuscript.

**2. Figures: The manuscript still has too many figures. I suggest combining some, like Figures 8 and 9, Figures 19 and 20, etc.**

**RESPONSE:** Thank you very much for your suggestion. We have carefully reviewed the figures and made adjustments as per your recommendation. Specifically, we have combined Figures 8 and 9 into a

single figure (now Figure 8) and Figures 19 and 20 into another single figure (now Figure 19). The adjusted figures are as follows:

[Figure]

**Figure 8. 2023 East Asia 10 m resolution rice distribution map (EARice10) and statistical analysis of rice area in different geographical regions: (a) full coverage of EARice10; (b)-(h) zoomed views of rice distribution in selected regions: (b) Xinjiang, China (provincial rice planting area less than 100,000 ha); (c) Heilongjiang, China (single-season rice region); (d) Hunan, China (mixed-season rice region); (e) Guangdong, China (double-season rice region); (f) the Democratic People's Republic of Korea (single-season rice region); (g) the Republic of Korea (single-season rice region); (h) Japan (single-season rice region); (i)-(l) Statistical analysis of rice area in different geographical regions: (i) Longitude; (j) Latitude; (k) DEM; (l) Slope.**

[Figure]

**Figure 18.    Number of cloud-free semi-monthly pixels from 2020 to 2023 and proportion of cloud-free semi-monthly pixel counts in different countries: (a)-(d) Number of cloud-free semi-monthly pixels from 2020 to 2023, where (a), (b), (c), and (d) represent 2020, 2021, 2022, and 2023, respectively; (e)-(h) Mean proportion of cloud-free semi-monthly pixel counts from 2020 to 2023 in different countries: (e) China, (f) Japan, (g) Republic of Korea, (h) Democratic People's Republic of Korea.**

**3. Figures 10-12: the official statistical data can be added to the bar charts for comparison.**

**RESPONSE:** Thank you very much for your suggestion. We have added the official statistical data to the bar charts in Figures 9−12 in the manuscript to facilitate comparison. The updated figures are as follows:

[Figure]

**Figure 9. Ten-meter rice distribution map in China and provincial rice area statistics (2023).**

[Figure]

**Figure 10. Ten-meter rice distribution map in Japan and provincial rice area statistics (2023).**

[Figure]

**Figure 11. Ten-meter rice distribution map in the Republic of Korea and provincial rice area statistics (2023).**

[Figure]

**Figure 12. Ten-meter rice distribution map in the Democratic People's Republic of Korea and provincial rice area statistics (2023).**

**4. Line 303-308: these statements can be removed as well.**

**RESPONSE:** Thank you very much for your suggestion. Following your recommendation, we have removed the statements in Lines 303–308 of the original manuscript to enhance conciseness.

**Response to Reviewer #4**

**Comments to the Author**

**The manuscript has been greatly improved and is now in relatively good shape. I have a few minor questions.**

**RESPONSE:** Thank you very much for your appreciation of our work. We sincerely appreciate your valuable suggestions, which have greatly contributed to enhancing the quality of our manuscript.

**1. The font size in the figures is too small, particularly in scatter plots like Figure 15 and Figure 18. What does the "450" under Figure 15c represent?**

**RESPONSE:** Thank you very much for your suggestion. Following your recommendation, we have redrawn Figures 15 and 18 from the original manuscript (now Figures 14 and 17 in the revised manuscript) and enlarged the font size to improve readability.

The number "450" under Figure 15c in the original manuscript was mistakenly introduced during the compilation of the four subfigures into a single figure. It holds no specific meaning and has been removed.

Once again, we sincerely appreciate your valuable feedback. The revised figures are as follows:

[Figure]

**Figure 14. Comparison of the extracted rice area from the EARice10 with the rice area from statistical yearbooks at the administrative division scale: (a) municipal-level comparison in China; (b), (c), and (d) provincial-level comparisons in Japan, the Republic of Korea, and the Democratic People's Republic of Korea, respectively.**

[Figure]

**Figure 17. Comparison between EARice10 and existing datasets from different countries' administrative regions.**

**2. It was reported that direct seeding is adopted in rice paddies in East Asia. Does the model capture such areas?**

**RESPONSE:** Thank you very much for your comment. Our proposed model is capable of accurately identifying areas of direct-seeded paddy rice cultivation, as explained in detail below.

In East Asia, rice is primarily grown in paddy fields, with transplantation and direct seeding being the most common cultivation methods. Unlike transplantation, direct seeding omits the process of transplanting seedlings. However, both methods share a common feature: flooding during the early stages of rice cultivation (Guo et al., 2019; Li et al., 2019). By detecting flooding signals during the early stages of rice growth, which typically correspond to the sowing and transplanting periods for transplanted rice, remote sensing imagery can effectively distinguish paddy rice fields from other land covers (Dong et al., 2016; Han et al., 2021; Han et al., 2022). The sowing and transplanting periods for transplanted rice span a relatively long period, during which flooding signals from concurrently planted direct-seeded rice can also be detected. Therefore, the sowing and transplanting window for transplanted rice can be utilized to simultaneously identify both transplanted and direct-seeded rice.

Our proposed method is based on the early-stage flooding signals characteristic of paddy rice. Specifically, for optical data, Sentinel-2 imagery was utilized to detect flooding signals during the sowing and transplanting periods using the LSWI and EVI indices, which were then used to generate optical-based rice candidate areas. These indices have been widely applied in large-scale rice mapping studies across Asia and have demonstrated high classification accuracy (Xiao et al., 2005; Xiao et al., 2006; Dong et al., 2016). Meanwhile, SAR data, with its all-time, all-weather operational capabilities, was combined with phenological information to achieve high-accuracy rice mapping independently (Zhan et al., 2021; Xu et al., 2023). By analyzing the minimum backscatter coefficient ($\sigma_{min}^0$), a key statistical parameter during the rice phenological period, flooding signals were effectively captured. Building on this parameter and integrating additional statistical features, we developed the SAR-based Rice Mapping Index (SRMI), a novel index specifically designed to identify SAR-based rice candidate areas.

To minimize uncertainties introduced by relying on a single data source, we integrated rice candidate areas derived from both optical and SAR data and implemented strict sample selection criteria to construct the training sample set (as described in Equation (9)). Subsequently, we developed and trained an optical-SAR adaptive fusion model by stacking multiple Random Forest classifiers to fully exploit the complementary strengths of SAR and optical data, resulting in a high-accuracy paddy rice distribution map. Using an independent validation sample set comprising 91,320 samples, we conducted a comprehensive evaluation of rice mapping accuracy across various regions in East Asia (as shown in Figure 13), confirming the reliability of our proposed method.

As shown in Figure 13, in provinces with higher proportions of direct-seeded rice, such as Zhejiang, Hubei, and Guangdong in China (Sha et al., 2019), the EARice10 dataset achieved user and overall accuracies exceeding 88%, highlighting the reliability of our method in identifying direct-seeded rice.

In conclusion, the proposed method effectively captures the distribution of paddy rice cultivated through both transplantation and direct seeding, offering a robust solution for high-accuracy paddy rice mapping.

[Figure]

Figure 13. Provincial-level confusion matrix metrics for the EARice10 based on validation sample set

**3. Regarding the validation in Figure 15, I believe the reported statistical data represent the planted area, which may differ slightly from the double-cropped area derived from the map. Did you consider the double-cropped area in the validation figure?**

RESPONSE: Thank you very much for your comment. We acknowledge that our previous description might not have been sufficiently detailed. In the validation figure, we have considered the planted areas of double-cropped rice. The statistical yearbooks report the planted areas for early, middle, and late rice, where middle rice corresponds to single-season rice, while early and late rice belong to double-season rice systems.

Since the rice area derived from the EARice10 map represents the annual rice distribution, for double-cropped rice regions, we used the maximum value between the planted areas of early and late rice as the official statistical data to ensure a reasonable comparison with the EARice10-derived rice area.

Thank you again for your suggestion. We have clarified this point in the manuscript as follows:

**Line 344, Page 21**

...Notably, the EARice10 reflects the annual distribution of rice, whereas the statistical yearbooks report the planted areas for early, middle, and late rice, with middle rice being single-season rice and early and late rice classified as double-season rice systems. Therefore, for double-season rice areas, we used the maximum value between early and late rice as the official statistical data to ensure a reasonable comparison...

**References**

Dong, J., Xiao, X., Menarguez, M. A., Zhang, G., Qin, Y., Thau, D., Biradar, C., and Moore, B., 3rd: Mapping paddy rice planting area in northeastern Asia with Landsat 8 images, phenology-based algorithm and Google Earth Engine, Remote Sens Environ, 185, 142-154, 10.1016/j.rse.2016.02.016, 2016.

Guo, Y. Q., Jia, X. P., Paull, D., and Benediktsson, J. A.: Nomination-favoured opinion pool for optical-SAR-synergistic rice mapping in face of weakened flooding signals, Isprs Journal of Photogrammetry and Remote Sensing, 155, 187-205, 10.1016/j.isprsjprs.2019.07.008, 2019.

Han, J., Zhang, Z., Luo, Y., Cao, J., Zhang, L., Cheng, F., Zhuang, H., Zhang, J., and Tao, F.: NESEA-Rice10: high-resolution annual paddy rice maps for Northeast and Southeast Asia from 2017 to 2019, Earth System Science Data, 13, 5969-5986, 10.5194/essd-13-5969-2021, 2021.

Han, J., Zhang, Z., Luo, Y., Cao, J., Zhang, L., Zhuang, H., Cheng, F., Zhang, J., and Tao, F.: Annual paddy rice planting area and cropping intensity datasets and their dynamics in the Asian monsoon region from 2000 to 2020, Agricultural Systems, 200, 10.1016/j.agsy.2022.103437, 2022.

Li, H., Guo, H. Q., Helbig, M., Dai, S. Q., Zhang, M. S., Zhao, M., Peng, C. H., Xiao, X. M., and Zhao, B.: Does direct-seeded rice decrease ecosystem-scale methane emissions?-A case study from a rice paddy in southeast China, Agricultural and Forest Meteorology, 272, 118-127, 10.1016/j.agrformet.2019.04.005, 2019.

Sha, W., Chen, F., and Mishra, A. K.: Adoption of direct seeded rice, land use and enterprise income: Evidence from Chinese rice producers, Land Use Policy, 83, 564-570, 10.1016/j.landusepol.2019.01.039, 2019.

Xiao, X., Boles, S., Frolking, S., Li, C., Babu, J. Y., Salas, W., and Moore III, B.: Mapping paddy rice agriculture in South and Southeast Asia using multi-temporal MODIS images, Remote sensing of Environment, 100, 95-113, 10.1016/j.rse.2005.10.004, 2006.

Xiao, X., Boles, S., Liu, J., Zhuang, D., Frolking, S., Li, C., Salas, W., and Moore III, B.: Mapping paddy rice agriculture in southern China using multi-temporal MODIS images, Remote sensing of environment, 95, 480-492, https://doi.org/10.1016/j.rse.2004.12.009, 2005.

Xu, S., Zhu, X., Chen, J., Zhu, X., Duan, M., Qiu, B., Wan, L., Tan, X., Xu, Y. N., and Cao, R.: A robust index to extract paddy fields in cloudy regions from SAR time series, Remote Sensing of Environment, 285, 10.1016/j.rse.2022.113374, 2023.

Zhan, P., Zhu, W., and Li, N.: An automated rice mapping method based on flooding signals in synthetic aperture radar time series, Remote Sensing of Environment, 252, 10.1016/j.rse.2020.112112, 2021.

---

## Author Response (AR3)

Dear Editorial Support Team,

*For the next revision, please check if your figures containing maps/aerial images require a copyright statement/image credit and add it to the figures (or captions) (https://publications.copernicus.org/for_authors/manuscript_preparation.html#mapsaerials). If these figures were entirely created by the authors, there is no need to add a copyright statement or credit. In that case it is important that you confirm this explicitly by email.*

**RESPONSE:**

Thank you for your notification and guidance regarding figures containing maps or aerial images.

We would like to confirm that, with the exception of the map base layer used in Figure 2-g, all other figures in our manuscript were entirely created by us and do not require copyright statements. For Figure 2-g, the required copyright statement for the map base layer has already been included in the submitted version, as follows:

*Basemap sources for (g): Esri, TomTom, Garmin, FAO, NOAA, USGS, © OpenStreetMap contributors, and the GIS User Community.*

Thank you again for your support. Please let us know if further clarification is needed.

Best regards,

Hong Zhang